



# The Actuator Farm Model for LES of Wind Farm-Induced Atmospheric Gravity Waves and Farm-Farm Interaction

Sebastiano Stipa[1], Arjun Ajay[1], and Joshua Brinkerhoff[1]

[1]School of Engineering, University of British Columbia–Okanagan, Kelowna, Canada

**Correspondence:** Sebastiano Stipa (sebstipa@mail.ubc.ca)

**Abstract.** This study introduces the actuator farm model (AFM), a novel parameterization for simulating wind turbines within large eddy simulations (LESs) of wind farms. Unlike conventional models like the actuator disk (AD) or actuator line (AL), the AFM utilizes a single actuator point at the rotor center and only requires 2-3 mesh cells across the rotor diameter. Turbine force is distributed to the surrounding cells using a new projection function characterized by an axisymmetric spatial support in the rotor plane and Gaussian decay in the streamwise direction. The spatial support's size is controlled by three parameters: the half-decay radius $r_{1/2}$, smoothness $s$, and streamwise standard deviation $\sigma$. Numerical experiments on an isolated NREL 5MW wind turbine demonstrate that selecting $r_{1/2} = R$ (where $R$ is the turbine radius), $s$ between 6 and 10, and $\sigma \approx \Delta x/1.6$ (where $\Delta x$ is the grid size in the streamwise direction) yields wake deficit profiles, turbine thrust, and power predictions similar to those obtained using the ADM, irrespective of horizontal grid spacing down to the order of the rotor radius.

Using these parameters, LESs of a small cluster of 25 turbines in both staggered and aligned layouts are conducted at different horizontal grid resolutions using the AFM. Results are compared against ADM simulations employing a spatial resolution that places at least 10 grid points across the rotor diameter. The wind farm is placed in a neutral atmospheric boundary layer (ABL) with turbulent inflow conditions interpolated from a previous simulation without turbines, referred to as a precursor. The implications of coarsening the grid are discussed for both the precursor and the wind farm simulation, and a new wall modelling approach is introduced that ensures a correct shear stress profile throughout the boundary layer, even when the grid resolution is too coarse to strictly guarantee law of the wall scaling. At horizontal resolutions finer than or equal to $R/2$, the AFM yields similar velocity, shear stress, turbine thrust and power as the ADM. Coarser resolutions reveal the AFM's ability to accurately capture power at the non-waked wind farm rows, although underestimating the power of waked turbines. However, the far wake of the cluster can be predicted well even when the cell size is of the order of the turbine radius.

Finally, combining AFM with a domain nesting method allows us to conduct simulations of two aligned wind farms in a fully-neutral ABL and of wind farm-induced atmospheric gravity waves under conventionally neutral ABL, obtaining excellent agreement with ADM simulations but with much lower computational cost. The simulations highlight the AFM's ability to investigate the mutual interactions between large turbine arrays and the thermally stratified atmosphere.



# 1 Introduction

The global offshore wind capacity has been expanding at a rapid rate in the past few years, with the global installed capacity estimated to reach 500 GW by the end of 2030 according to the 1.5°C global temperature rise scenario (IRENA, 2023). This signifies a fourteen-fold increase compared to the 2020 levels. As the number and size of offshore wind farms increases, they are often clustered to maximize use of the available wind energy resources as well as to minimize the infrastructure costs (Akhtar et al., 2021; Junqueira et al., 2021). Wind farm clusters can lead to reduced wind farm power production due to the impact of wakes shed by upstream wind farms, which may persist for several kilometers downstream as reported from airborne (Platis et al., 2018; Lampert et al., 2020), LiDAR (Bodini et al., 2021; Schneemann et al., 2020), satellite synthetic aperture radar (SAR) measurements (Hasager et al., 2015; Ahsbahs et al., 2020) as well as mesoscale numerical studies (Pryor and Barthelmie, 2024; Fischereit et al., 2022; Wang et al., 2023). The extent of wind farm wake propagation depends on factors such as atmospheric stability, turbulent intensity, and boundary layer height as well as wind farm size (Schneemann et al., 2020; Cañadillas et al., 2020; Pryor et al., 2021). Moreover, recent numerical studies (Stipa et al., 2024b; Maas and Raasch, 2022) have shown how wind-farm generated atmospheric gravity waves can be triggered within a shallow boundary layer under stable free atmosphere stratification, leading to horizontal pressure gradients that influence the flow in the region upstream of the farm — leading to what is commonly referred to as blockage — as well as the wind farm wake. This underlines the importance of studying the mutual interactions between neighbouring farms as well the interactions of farm clusters with the atmospheric boundary layer (ABL) and the free atmosphere above.

Wind farm clusters have been studied using various numerical models differing in the breadth of the resolved temporal and spatial scales. These include engineering models, computational fluid dynamics (CFD) models such as large eddy simulation (LES) or Reynolds averaged Navier-Stokes (RANS), or high-fidelity mesoscale numerical models such as the Weather Research and Forecasting (WRF) model (Skamarock et al., 2019).

Engineering models usually combine different sub-models to capture various physical processes such as individual turbine wakes (Bastankhah and Porté-Agel, 2014; Blondel and Cathelain, 2020), their interaction and merging (Niayifar and Porté-Agel, 2016) and blockage effects (Branlard and Meyer Forsting, 2020). On the one hand, turbine wake models tend to underestimate wind farm wake loss (Fischereit et al., 2022; van der Laan et al., 2023b) due to the assumptions regarding the wake profile, wake expansion and superposition of wakes. Additionally, Gayle Nygaard et al. (2020) showed that individual wake models strongly overestimate wind farm wake recovery when applied to large clusters. On the other hand, individual blockage models underestimate the full extent of the blockage effect as they do not consider the wind farm interaction with the atmosphere. In this regard, Stipa et al. (2024b) and Devesse et al. (2023) showed that the accuracy of these engineering models improves substantially when they are coupled with a reduced-order mesoscale model, making them more suitable when looking at large wind farm clusters.

High-fidelity mesoscale models like WRF utilize a coarse grid resolution, such that the flow around individual wind turbines is not resolved and hence the wind farm drag force must be parameterized. The Fitch et al. (2012) wind farm parametrization model represents the effect of wind turbines as a momentum sink. Specifically, a portion of the flow kinetic energy is used to



create electricity while the rest is converted into turbulent kinetic energy (TKE). Eriksson et al. (2015) simulated the Lillgrund wind farm, located off the coast of southern Sweden, and compared WRF's wind farm parametrizations against large eddy simulation (LES) data, where wind turbines were modeled using the actuator disk method (Mikkelsen, 2003). In this analysis, the WRF model overestimated the wind farm power and predicted faster wake recovery compared to LES. Studies by Vanderwende et al. (2016); Peña et al. (2022) also indicate that adopting WRF's turbine parameterizations yields lower velocity deficits and higher TKE in the wind turbine wake than a comparable LES. This suggests that, when possible, mesoscale models require validation against microscale models, which are capable of resolving finer temporal and spatial scales.

In recent years, there has been a significant increase in studies employing LES to examine the flow around large wind farms and the evolution of cluster wake (see for example Maas, 2023; Maas and Raasch, 2022; Cheung et al., 2023; Stipa et al., 2024a; Lanzilao and Meyers, 2022b; Stieren and Stevens, 2022, among others). This trend is supported by the growing availability of computational resources, allowing the temporal fluctuations and large-scale flow features of the ABL to be captured alongside the dynamics of wind turbine wakes, which are both important for accurate simulation of wind farm clusters. Nevertheless, these simulations still require significant computational resources due to the size of the simulation domain and the grid resolution constraints. For instance, Stieren and Stevens (2022) used LES to study the impact of a rectangular wind farm wake on another wind farm located downstream in fully-neutral ABL conditions, with a horizontal domain size of about $390 \text{ km}^2$ and $1800 \times 480 \times 480$ mesh cells. To simulate real-world wind farm clusters, an even larger domain is usually required. Maas and Raasch (2022) performed a series of LES of the wind farm clusters in the German Bight under different ABL stability conditions, using a horizontal domain size of approximately $33\,620 \text{ km}^2$, around 8.4 billion mesh cells and finest grid spacing of $20 \times 20 \times 20$ m. (Cheung et al., 2023) simulated the $10\,000 \text{ km}^2$ AWAKEN wind farms site in Oklahoma (USA) using 21.4 billion mesh cells, a background mesh size of $20 \times 20 \times 20$ m and finest grid spacing of $2.5 \times 2.5 \times 2.5$ m. Additionally, the vertical extent of the domain is also increased when simulating the effects of wind farm induced gravity waves within the free atmosphere due to the large vertical wavelength of these waves (Allaerts and Meyers, 2017)

Wind farm LES studies typically employ a precursor-successor method, where the precursor simulation is used to develop the ABL turbulent inflow which is subsequently utilized in the successor simulation which includes the wind turbines. In the context of ABL LES, the grid resolution is dictated by the requirement to correctly capture the theoretical law of the wall (LOTW) scaling, which imposes a specific cell aspect ratio at the wall, as well as a minimum number of cells within the boundary layer (Brasseur and Wei, 2010). Furthermore, the inclusion of wind turbines in the successor simulation typically introduces additional grid resolution constraints which depend on how wind turbines are represented within the numerical domain. The actuator line model (ALM, Sørensen and Shen, 2002) is usually employed for simulations of isolated wind turbines or small clusters of 2-3 machines due to its requirement that the rotating blade tip cannot cross more than one mesh cell per time step, which substantially limits the time step size. For this reason, LES studies of large wind farms usually employ the actuator disk model (ADM, Mikkelsen, 2003), which requires at least 8 grid points across the turbine diameter for sufficient spatial resolution (Wu and Porté-Agel, 2013) and leaves the time step to be determined according to the flow solution. This restricts the grid size in the range of 15 m - 30 m for most LES studies, depending on the specific wind turbine diameter. Notably, those studies that aim at investigating the effect of wind direction changes (see for example Stieren et al., 2021)





require to maintain this condition not only in the spanwise and vertical direction, but also in the streamwise, as turbines or the wind rotate dynamically.

Within this landscape, it is evident that alternate wind turbine models that overcome the grid resolution constraint introduced by existing actuator models will be beneficial to reduce the computational cost of conducting LESs of large wind farms, especially when looking at farm-farm interactions. Recently, van der Laan et al. (2023a) developed a RANS-based wind farm parametrization model similar to a forest canopy model that applies a wind farm drag obtained by filtering each wind turbine location using a two-dimensional Gaussian function. While the model compares well against RANS-ADM simulations when

looking at the entire cluster, it requires multiple RANS-ADM simulations of every represented wind farm in a study in order to compute the wind-drag coefficient relation, which is required as a model input.

In the present work, a new actuator model referred to as the actuator farm model (AFM), is developed and validated. In contrast to conventional actuator models, the AFM requires a single actuator point positioned at the rotor center and only 3-4 mesh cells across the rotor diameter. This essentially reduces the grid constraint only to that imposed by the simulation's

ability to capture law-of-the-wall scaling. The turbine force is projected from the actuator point to the surrounding grid cells using a new projection function characterized by axisymmetric spatial support in the rotor plane and Gaussian decay in the streamwise direction. Although the AFM solution of a single turbine simulation approaches the ADM solution when a similar grid size is used, the application domain of the AFM is tailored to problems requiring large domains that would otherwise be too computationally expensive, such as studies of cluster wake evolution, farm-farm and farm-ABL interactions. In terms of

model fidelity at the turbine scale, AFM-LES lies in between the more detailed ADM-LES and the parameterizations employed in numerical weather prediction codes (e.g. Fitch et al., 2012). Two illustrative applications of the AFM are the investigation of wind-farm induced atmospheric gravity waves and farm-farm interaction via a hybrid AFM-ADM approach, whereby an upstream wind farm is modeled using AFM with a coarser grid, while a downwind farm of interest is represented using ADM within a nested finer grid.

The present work is organized as follows. Section 2 introduces the AFM, the grid nesting method used in the present study, and considerations regarding the effect of coarsening the grid size in the context of wind farm LES. Section 3 presents the parametric analysis conducted on an isolated wind turbine to choose the best set of AFM parameters and to investigate the sensitivity of the model to the grid size. Section 4 describes the AFM simulations performed on both an aligned and a staggered wind farm layout, showing their comparison against ADM results. In Sections 5 and 6, the AFM combined with grid nesting is

leveraged to study the interaction between two aligned wind farms and to simulate wind-farm induced gravity waves. Finally, Section 7 highlights the conclusions of the present study.

## 2    Methodology

For the LES simulations presented in this paper, we use the open-source finite volume code TOSCA (Toolbox fOr Stratified Convective Atmospheres) developed at the University of British Columbia. The governing equations correspond to mass and

momentum conservation, while sub-grid-scale stresses are calculated with the model proposed by Meneveau et al. (1996),



where the dynamic Smagorinsky model coefficient is averaged along the flow pathlines in a Lagrangian sense. A potential temperature transport equation can be also solved to account for stability effects inside the ABL and in the free atmosphere. As specific details about TOSCA's numerical method are provided together with an exhaustive validation in Stipa et al. (2024b), the present section focuses only on new features that are relevant for the current paper. Specifically, the conventional non-rotating uniform ADM is described in Section 2.1, while Section 2.2 presents the developed AFM. Lastly, Section 2.5 describes the domain nesting technique used in TOSCA, referred to as overset mesh, which will be used for the simulations described in Sections 5 and 6.

## 2.1 Uniform Actuator Disk Model

The ADM represents each individual wind turbine within the computational domain by discretizing each rotor disk with a certain number of actuator points in the radial and azimuthal directions. In principle, the ADM allows to model a radially-varying blade force upon knowing the blade chord, twist and type of airfoil at each radial location and by providing lift and drag look-up tables for each airfoil (Martínez-Tossas et al., 2015a). In this case, both wind turbine thrust and torque are modeled and the wind turbine can be additionally equipped with angular velocity and pitch controllers. However, thrust and power coefficients are computed variables that vary in time for this class of ADM, making it more difficult to compare their results against other classes of ADM where the turbine thrust coefficient is a model input (the so-called "uniform" ADM; Porté-Agel et al., 2010; Calaf et al., 2010; Jimenez et al., 2007, 2008). As a consequence, in order to exactly match the turbine thrust coefficient applied in the ADM to that of the AFM when comparing the two models, we employ the uniform ADM throughout the entire manuscript. This class of ADMs only applies a thrust force to the flow, making it unable to capture the tangential force exerted on the flow by the wind turbine. As a consequence, uniform ADMs are only expected to produce accurate results in the far region of the wind turbine wake. For brevity, the modifier uniform will be omitted throughout the remainder of the manuscript.

Wind turbine thrust at each actuator point is calculated as

$$\boldsymbol{T}_p = -\frac{1}{2}||\boldsymbol{U}_d||\boldsymbol{U}_d C'_T dA_p, \tag{1}$$

where $dA_p$ is the portion of rotor disk area associated to each actuator point and $\boldsymbol{U}_d$ is the disk velocity sampled from the background mesh at the actuator points, averaged among all actuator points; the computation of the disk velocity $\boldsymbol{U}_d$ is defined in section 2.2. The disk-based thrust coefficient $C'_T$ can be determined from the thrust coefficient $C_T$ using the relation

$$C'_T = \frac{C_T}{(1-a)^2}, \tag{2}$$

where $a$ is the turbine axial induction factor (Calaf et al., 2010). The point force at each actuator point should be projected to the mesh cells and transformed into a body force field that represents the effect of the wind turbine on the incoming flow. This operation is performed by means of the projection kernel $g_{AD}(\boldsymbol{x})$, defined as

$$g_{AD}(\boldsymbol{x}) = \frac{1}{\epsilon^3 \pi^{3/2}} \exp\left(-\frac{(x_c-x_p)^2}{\epsilon^2} - \frac{(y_c-y_p)^2}{\epsilon^2} - \frac{(z_c-z_p)^2}{\epsilon^2}\right), \tag{3}$$



where subscripts $c$ and $p$ refer to the mesh cell center and actuator point, respectively. The quantity $\epsilon$ is the projection width and it should be set to $1.5 - 2$ times the mesh size along the rotor plane (Calaf et al., 2010; Martínez-Tossas et al., 2015b). Using Equation (3), the wind turbine body force at each mesh cell can be evaluated as

$$\boldsymbol{b}_c = \sum_{p=1}^{N_p} g_{AD} \boldsymbol{T}_p. \tag{4}$$

where $N_p$ is the total number of actuator points. Note that the body force contribution at any given cell $c$ may come from different actuator points. Moreover, the sum of $\boldsymbol{T}_p$ among all actuator points yields a force that is equivalent to the total wind turbine thrust but directed in the opposite direction.

## 2.2 Actuator Farm Model

The AFM is based on a similar concept to that of the ADM but, instead of representing each wind turbine as a distribution of actuator points, a single point is used, located at the rotor center. Hence, the thrust force at this single actuator point coincides with the total wind turbine thrust and it is evaluated as

$$\boldsymbol{T} = -\frac{1}{2}||\boldsymbol{U}_d||\boldsymbol{U}_d C_T' \pi R^2, \tag{5}$$

where $R$ is the turbine radius. The main implication of using a single actuator point is that the spatial support of the body force field, given by the projection kernel convolution with the actuator point locations in the ADM, is equivalent to the projection kernel itself in the AFM. For this reason, the spatial support of the projection function should be of the order of the rotor disk. One option is to use the Gaussian kernel expressed by Equation (3), with $\sigma \geq R$, to distribute the turbine thrust to the neighboring mesh cells. However, this approach produces a body force that is too concentrated close to the rotor center (not shown here), resulting in an artificial shedding of the wind turbine wake for high thrust coefficients. For this reason, a new projection function has been developed which emulates the one resulting from the convolution of Equation (3) with the actuator disk point locations. In addition, any kernel used within actuator models should in general integrate to unity over the volume so that the total wind turbine thrust is recovered upon integrating the body force.

First, we define a Cartesian coordinate system $\mathcal{C}$, having its origin at the rotor center, the $x$ axis aligned with the wind direction, the $z$ axis pointing in the vertical direction and the $y$ axis to form a right-handed coordinate frame. Within $\mathcal{C}$, we can define a function $f(\boldsymbol{x})$ as

$$f(\boldsymbol{x}) = \frac{1}{\exp\left(\frac{\sqrt{y^2+z^2}-r_{1/2}}{s}\right)+1} \exp\left(-\frac{x^2}{\sigma^2}\right), \tag{6}$$

where $r_{1/2}$ is the radius in the $(y,z)$ plane where the function decays by $1/2$, $s$ is a smoothing parameter, and $\sigma$ is the standard deviation of the Gaussian decay along $x$. Equation (6) is axisymmetric on the rotor plane with respect to the rotor center, and it coincides with a Gaussian function in the $x$ direction. To be used as a projection kernel, it is divided by its integral over the volume so that the integral of the resulting function equals unity. In order to more easily perform the integral, Equation (6) is





expressed in cylindrical coordinates. As a result, expressing the differential volume given by $dxdydz$ as $rdrd\theta dx$, the definite integral of Equation (6) over the entire domain can be written as

$$
\mathcal{I} = \int_0^{2\pi} d\theta \int_{-\infty}^{\infty} \exp\left(-\frac{x^2}{\sigma^2}\right) dx \int_0^{\infty} \frac{r}{\exp\left(\frac{r-r_{1/2}}{s}\right)+1} dr
$$
$$
= 2\sigma\pi^{3/2} \left[ s^2 Li_2\left(-\exp\left(\frac{r_{1/2}-r}{s}\right)\right) - sr\log\left(\exp\left(\frac{r_{1/2}-r}{s}\right)+1\right) \right]_0^{\infty}, \tag{7}
$$

where $Li_2$ is the poly-logarithmic function of the second kind and the expression between square brackets is the integral of the portion of Equation (6) which depends on $r$. Its values at zero and infinity are evaluated as follows

$$
\left[ s^2 Li_2\left(-\exp\left(\frac{r_{1/2}-r}{s}\right)\right) - sr\log\left(\exp\left(\frac{r_{1/2}-r}{s}\right)+1\right) \right]_0 = s^2 Li_2\left(-\exp\left(\frac{r_{1/2}}{s}\right)\right), \tag{8}
$$

$$
\lim_{r\to\infty} \left[ s^2 Li_2\left(-\exp\left(\frac{r_{1/2}-r}{s}\right)\right) - sr\log\left(\exp\left(\frac{r_{1/2}-r}{s}\right)+1\right) \right] = 0. \tag{9}
$$

Hence, the functional form of the projection function which integrates to unity is finally given by

$$
g_{AF}(x,r) = -\frac{1}{2\sigma\pi^{3/2}s^2 Li_2\left[-\exp\left(\frac{r_{1/2}}{s}\right)\right]} \left( \frac{\exp\left(-\frac{x^2}{\sigma^2}\right)}{\exp\left(\frac{r-r_{1/2}}{s}\right)+1} \right), \tag{10}
$$

where a minus sign has been applied to Equation (8) due to the integration in Equation (7). Using Equation (10), the wind turbine body force at each mesh cell can be evaluated as $\boldsymbol{b}_c = g_{AF}\boldsymbol{T}$. Notably, Equation (10) has two free parameters, namely the half-decay radius $r_{1/2}$ and the smoothing $s$, while the streamwise standard deviation $\sigma$ can be chosen following the same approach as the ADM.

As an illustrative example, the wind turbine force calculated using $\boldsymbol{U}_d = 9$ m/s, $R = 60$ m and $C'_T = 0.7$ is shown in Figure 1 for both the ADM and AFM on a vertical plane through the turbine rotor center. The grid size along $y$ and $z$ is set to $5 \times 5$ m, $\sigma$ is set to 20 m for both Equation (3) and Equation (10), while $r_{1/2}$ and $s$ are set to 60 m and 6, respectively. As can be appreciated, the definition of $g_{AF}$ allows for the recovery of a body force field that is similar to that obtained from the ADM. However, this is achieved in the AFM using a single function that projects from the wind turbine rotor center, instead of the

summation of many different Gaussian functions centered at each actuator point. This property allows the AFM to require fewer mesh cells along the wind turbine radius to properly resolve the projection function in space.

The developed AFM projection function is limited by the wind turbine hub-height to avoid error in the body force owing to some of the force being projected outside of the domain. In practice, the $r_{1/2}$ parameter should be of the order of the wind turbine radius. Figure 2 shows the radially-dependent component of Equation (10) for two values of $r_{1/2}$ and different values of

smoothing parameter $s$. The vertical dashed line indicates the wall, located at $h_{hub}/R = 1.5$ from the rotor center. In order not to incur in any projection error, Equation (10) should reach a value close to zero before reaching the edge of the domain. For





this reason, high values of $s$ are not ideal, as they would yield a defect in the recovered wind turbine force following integration over the domain.

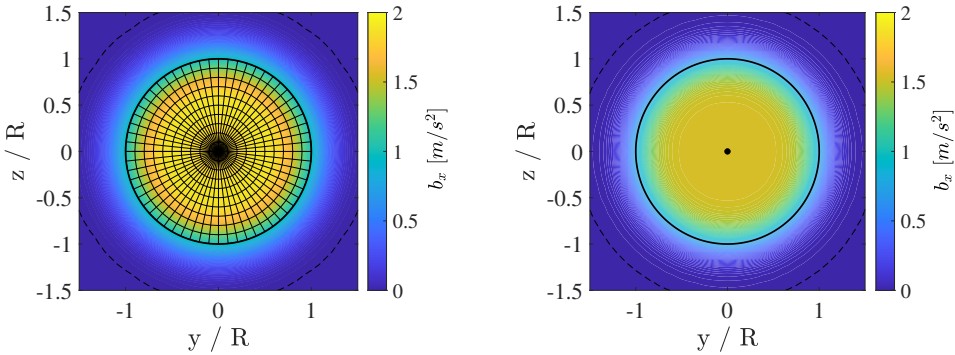

**Figure 1.** Comparison of two sections located at $x = 0$ of the body force projection function for the ADM (left) and AFM (right). In the ADM, the force associated to each actuator point is equal to the total wind turbine force scaled by the ratio between the actuator element area and the rotor swept area. For the AFM, the force is projected from a single actuator point, located at the rotor center. The black continuous circle indicates the radial coordinate corresponding to the wind turbine radius. The parameter $\sigma$ is set to 20 m for both Equation (3) and Equation (10), while $r_{1/2}$ and $s$ are set to 60 m and 6, respectively.

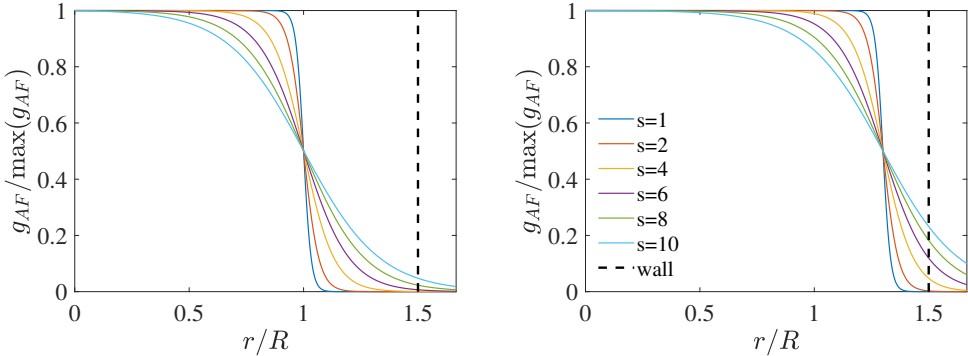

**Figure 2.** Dependency of Equation (10) on the parameters $r_{1/2}$ (left: $r_{1/2}/R = 1$, right: $r_{1/2}/R = 1.3$) and on the smoothing $s$. The vertical dashed line indicates the wall, located at a distance $h_{\text{hub}}$ from the rotor center ($h_{\text{hub}}/R = 1.5$ in the figure, but this ratio depends on the specific wind turbine under study).

Regarding the calculation of the disk velocity $\boldsymbol{U}_d$, two strategies have been tested within the present paper. A first method, referred to as the rotor disk sampling, consists of tri-linearly interpolating the wind speed at the rotor center from the 8 cells surrounding the actuator point. In a second strategy, inspired by Churchfield et al. (2017) and referred to as the integral





sampling, the disk velocity is calculated as

$$\boldsymbol{U}_d = \sum_{c=1}^{N_c} g_{AF} \boldsymbol{u}_c \tag{11}$$

where $\boldsymbol{u}_c$ is the wind speed at cell $c$ and $N_c$ is the number of cells contained in a sphere of radius $2r_{1/2}$ from the rotor center.

Notably, this radius is only defined for implementation purposes and increasing it further has no effect as $g_{AF} = 0$ decays to zero well before $2r_{1/2}$.

## 2.3 Effect of Spatial Resolution

One of the main benefits of the AFM is to relax the requirement imposed by the ADM which dictates that at least $8 - 10$ mesh cells should be used along the rotor diameter. This allows to save computational resources by reducing the number of cells in

the domain, especially for those wind farm simulations characterized by a domain that extends for tens of kilometers in each direction. However, while these large simulations are the main target for the AFM, it is crucial to understand the limits and implications of reducing the LES spatial resolution below what is commonly employed within the research community.

Inlet boundary conditions in wind farm LES are often calculated by means of a different simulation, referred to as the precursor. The precursor does not contain any wind turbine and generally employs periodic boundary conditions in the lateral

directions. This allows to recycle the flow for several flow-turnover times, until a fully developed ABL characterized by stationary turbulence statistics is reached. For non-idealized or time-varying atmospheric states corresponding to a specific realization of the planetary boundary layer, the precursor simulation can be forced using profile assimilation techniques or two-dimensional boundary data derived from weather models (see Haupt et al., 2023 for a review). The precursor simulation is then used to derive boundary conditions that characterize the incoming flow for the wind farm simulation, referred to as the

successor. In its most simple form, two-dimensional sections at a given streamwise coordinate are saved during the precursor at each time step and then used as inlet boundary condition for the successor simulation. If free atmosphere stratification is present and atmospheric gravity waves should be resolved, the precursor can be synchronized with the successor and used to prescribe the inlet flow through momentum and temperature source terms applied throughout a fringe region (see Allaerts and Meyers, 2017; Stipa et al., 2024b; Lanzilao and Meyers, 2022a, among others). More complex ways of forcing a successor

simulation involve the use of one- or two-way boundary-coupled nested domains such as in WRF-LES (Sanchez Gomez et al., 2022, 2023). In all these cases, the inflow data used to set the inlet boundary condition for the successor should exhibit the correct law of the wall (LOTW) scaling in the velocity profile, as well as the expected shear stress profile and friction velocity for the specific simulation conditions. In addition, the inflow data should be mapped to the wind farm simulation without being altered by the mapping procedure. In the reminder of this section, the effects of grid coarsening (both in the precursor and

successor) on these aspects will be addressed in detail.





Regarding the compliance with LOTW scaling and the recovery of representative shear stress profile and friction velocity, Brasseur and Wei (2010) identified three criteria that should be satisfied when running ABL simulations:

$$\mathcal{R}/\mathcal{R}^* > 1, \tag{12}$$

$$Re_{LES}/Re_{LES}^* > 1, \tag{13}$$

$$N_\delta/N_\delta^* > 1, \tag{14}$$

where $N_\delta$ is the number of cells used to resolve the boundary layer, $Re_{LES}$ is a Reynolds number calculated with a spurious length scale $\delta_{LES}$ arising due to spurious frictional forces in the sub-grid scale (SGS) model, and $\mathcal{R}$ is the ratio between the fluctuating resolved stresses and the modeled SGS stresses at the first cell center. The critical values, identified with $^*$, roughly correspond for the neutral ABL to $\mathcal{R}^* = 1$, $Re_{LES}^* \approx 300$ and $N_\delta^* \approx 50$. While the above criteria depend on the specific SGS closure employed within the LES, for an eddy-viscosity closure that employs the Smagorinsky model

$$\mathcal{R} \approx \frac{(N_\delta - 1)\kappa^2}{1.05 N_\delta C_s^2 A_R^{4/3}} \tag{15}$$

$$Re_{LES} \approx \frac{N_\delta}{\kappa}(\mathcal{R} + 1) \tag{16}$$

where $C_s$ is the Smagorinsky constant, $\kappa$ is the von Kármán constant and $A_R = \Delta h/\Delta z$ is the cell aspect ratio at the wall, with $\Delta h = \max(\Delta x, \Delta y)$. The criteria expressed by Equations (12) to (14), calculated assuming an eddy-viscosity closure using the Smagorinsky SGS model with $C_s = 0.1$ and a boundary layer height of $H = 750$ m are summarized in Table 1 for different values of the grid spacing. Notably, the quantity $\mathcal{R}/\mathcal{R}^*$ strongly depends on the employed model coefficient and the cell aspect ratio at the wall. As a rule of thumb, to improve LOTW scaling, one should increase the horizontal resolution while keeping the vertical resolution constant or reduce the model coefficient. The criteria expressed by Equations (12) to (14) can be used to estimate if the LES setup is suitable to capture LOTW scaling and adhering to these criteria is advisable, particularly when considering the ABL flow. However, numerous studies exist, mainly focused on the wind farm flow, wherein strict adherence to the Brasseur and Wei (2010) criteria is not observed, (as seen in Stieren and Stevens, 2022; Stipa et al., 2023a; Lanzilao and Meyers, 2023) but which still offer valuable insight regarding the underlying flow physics. Moreover, it should be noted that while failing to respect Brasseur and Wei (2010) criteria in the precursor simulation may lead to a consistent LOTW mismatch, as the flow is recycled multiple times over the domain, doing so in the successor wind farm simulation only leads to a potential mismatch that develops from the inlet, where the provided inflow is prescribed, to the outlet, where the wind has evolved according to the specific simulation setup. In general, the impact of satisfying Equations (12) to (14) only in the precursor and not in the wind farm simulation depends on additional factors, such as the size of the computational domain over which the mismatch accumulates as well as the employed SGS closure and numerical schemes. Moreover, it may also depend on the adopted simulation code.





| $\Delta x \times \Delta y \times \Delta z$ | $\mathcal{R}/\mathcal{R}^*$ | $Re_{LES}/Re_{LES}^*$ | $N_\delta/N_\delta^*$ |
|---|---|---|---|
| $15 \times 15 \times 10$ | 7.87 | 5.55 | 1.50 |
| $20 \times 20 \times 20$ | 14.24 | 4.76 | 0.75 |
| $30 \times 15 \times 5$ | 0.38 | 1.75 | 3.00 |
| $40 \times 16 \times 10$ | 1.40 | 1.50 | 1.50 |
| $50 \times 50 \times 10$ | 0.78 | 1.11 | 1.50 |
| $50 \times 50 \times 30$ | 6.71 | 1.61 | 0.50 |

**Table 1.** Brasseur and Wei (2010) criteria for different precursor mesh sizes, calculated assuming Smagorinsky model coefficient $C_s = 0.1$ and ABL height $H = 750$ m (these are representative values for wind farm LES). Critical values are set to $\mathcal{R}^* = 1$, $Re_{LES}^* = 300$ and $N_\delta^* = 50$.

In general, mapping the precursor inflow data to the successor domain inlet requires interpolation in both space and time, as the precursor and successor may not have the same mesh at the inlet nor have advanced with an identical time step size. When this is the case, the two-dimensional inflow data mapped at the successor inlet loses the property of being divergence free. Consequently, when dealing with an incompressible code, non-solenoidal fluctuations in velocity arising from the mapping will be advected into the internal cells, ultimately modifying the incoming profile of the resolved Reynolds stresses. The result is an imbalance in the momentum equation wherein the driving pressure gradient is no longer balanced by the resolved shear stress, causing the mean flow in the successor to accelerate or decelerate from the mapped inlet plane depending on whether the error in the resolved stress is positive or negative. We observe this behavior to be more prominent when interpolating from a finer to coarser mesh — i.e. when mapping yields a loss of information — and when the difference in mesh size is more than a factor of 2. To avoid this problem, continuity-preserving B-spline interpolation proposed by Schroeder et al., 2022 can be used for mapping inlet flow data instead of the classic bi-linear interpolation. However, we have found that this only yields marginal improvement, as spatial interpolation along $x$ is required to render the interpolated flow truly divergence-free and B-spline interpolation along $x$ requires saving three flow sections per time step, tripling the I/O overhead associated with precursor-successor mapping.

Based on the above discussion, it might be concluded that precursor and successor solutions should be conducted with identical spatial and temporal resolution to avoid altering the successor result by the inlet mapping procedure. However, a resolution that sufficiently captures the ABL turbulence in the precursor might be very restrictive for a successor simulation of a very large wind farm in terms of computational cost. This is the case of Maas (2023) and Cheung et al. (2023), who both used a grid size of $20 \times 20 \times 20$ m for their precursors and successor analyses. Such resolution is expected to capture the LOTW according to Equations (12) to (14), but led to $6.8$ and $21.14$ billion mesh elements, respectively (Cheung et al. (2023) additionally used mesh refinement around the wind turbines), making these wind farm simulations perhaps the largest conducted to-date. The Brasseur and Wei (2010) criteria were also satisfied for the LES performed by Wu and Porté-Agel (2017), who used a grid spacing of $40 \times 16 \times 10$ m for their precursor and successor simulations. An alternative approach





is to conduct the precursor using a coarser mesh, coincident with the lowest resolution required by the ADM to resolve the wind turbines. This is the case of Stieren and Stevens (2022) and Lanzilao and Meyers (2023), who used $30 \times 15 \times 5$ m and
$31.25 \times 21.74 \times 5$ m, respectively. Stipa et al. (2023a) conducted wind farm simulations using a $30 \times 15 \times 10$ m grid spacing, but precursors were carried out on a domain characterized by a $15 \times 15 \times 10$ m cells. While these values may not strictly adhere to the criteria defined by Brasseur and Wei (2010), they are reasonable approximations. Any potential LOTW mismatch is expected to be minimal and unlikely to significantly alter the simulation results with respect to a fully LOTW-compliant LES.

If the AFM is employed, the successor mesh can be further coarsened up to a grid spacing on the order of $40 - 60$ m in
the horizontal directions. In this case, interpolating the inflow data from a precursor that satisfies Brasseur and Wei (2010) criteria may lead to the alteration of the shear stress profile described above. On the other hand, when a similar resolution to the successor mesh is employed in the precursor domain, the LOTW mismatch becomes large, especially at the first cell center, potentially affecting the wall shear stress if a wall model is used. In fact, widely adopted wall models based on the classic Monin and Obukhov (1954) similarity theory compute the shear stress at the wall by applying the LOTW locally at the first
cell center. The velocity at the first cell is used to compute the friction velocity $u^*$ as

$$u^* = \frac{\kappa \sqrt{u_1^2 + v_1^2}}{\ln\left(\frac{z_1}{z_0}\right)}, \tag{17}$$

for the neutral ABL, where subscript 1 indicates quantities evaluated at the first cell center and $z_0$ is the equivalent roughness height. The wall shear stress is then calculated as

$$\tau_{xz}^{\mathrm{w}} = -u^{*2} \frac{u_1^2}{\sqrt{u_1^2 + v_1^2}}, \tag{18}$$

$$\tau_{yz}^{\mathrm{w}} = -u^{*2} \frac{v_1^2}{\sqrt{u_1^2 + v_1^2}}.. \tag{19}$$

From Equations (17) to (19), it is clear that a large LOTW mismatch, especially at the first cell, leads to an error in the wall shear stress and, in turn, in the vertical profile of resolved shear stress. While the LOTW mismatch only causes a departure of the wind profile from the logarithmic law of the wall, a mismatch in the shear stress profile is much more serious as it affects the turbulence intensity level experienced by the wind farm and possibly turbine and wind farm wake recovery. Hence, the wall
model has to be modified such that the correct wall shear stress is applied regardless of the employed grid resolution.

## 2.4   Numerical Setup

A precursor simulation that satisfies the criteria expressed by Equations (12) to (14) is first conducted on a fine grid and the resulting friction velocity $u_{\mathrm{fine}}^*$ is calculated using Equation (17). Then, a second precursor characterized by a coarser mesh where $\Delta x_{\mathrm{coarse}} > 2\Delta x_{\mathrm{fine}}$ and $\Delta y_{\mathrm{coarse}} > 2\Delta y_{\mathrm{fine}}$ is conducted and the wall model is modified such that Equations (18) and (19)
are used with $u^* = u_{\mathrm{fine}}^*$. This essentially renders the wall model independent of the employed grid size and ensures matching of the shear stress profile.

To verify this approach and to show the effects of coarsening the precursor simulation grid, we conducted simulations of a fully neutral atmospheric boundary layer using three different values of the grid spacing. The finer grid uses a resolution





of $15 \times 15 \times 10$ m, which is expected to satisfy all Brasseur and Wei (2010) criteria. The coarser cases use a resolution of

$50 \times 50 \times 10$ m and $50 \times 50 \times 30$ m, respectively. The former does not satisfy $\mathcal{R}/\mathcal{R}^* > 1$, while $Re_{LES}/Re_{LES}^* \approx 1$. However, a sufficient number of vertical grid nodes is used to resolve the ABL vertically. When the vertical grid spacing is increased, the first two criteria are satisfied, but $N_\delta/N_\delta^* < 1$. Both coarser cases are simulated using the conventional wall model and the modified wall model, where $u_{\text{fine}}^*$ is calculated from the precursor characterized by the finer grid.

As no stratification is present, the domain size is set to $4.2 \times 4.2 \times 0.7$ km, essentially fixing the ABL height $H$ to 700 m.

The Coriolis parameter $f_c$ is set to $1.184 \times 10^{-4}$ s$^{-1}$, corresponding to a latitude of $54.5$ deg. The equivalent roughness height $z_0$ is set to $0.001$ m. Horizontal boundaries are periodic, while a stress-free condition is applied at the upper boundary. At the bottom, wall shear stress is directly applied in the momentum equation using Equations (18) and (19), while velocity at the ghost nodes is set such that the wall normal gradient at the boundary face coincides with the one evaluated at the first cell center. This boundary condition allows the velocity to be ultimately determined by the amount of shear stress applied by the

wall model. Moreover, since the entire wall shear stress is modeled, the effective viscosity is set to zero at the wall to avoid double counting. A uniform driving pressure gradient is applied to the momentum equation such that the horizontally-averaged velocity at $h_{\text{ref}} = 90$ m is equal to 9 m/s (Stipa et al., 2024b). All simulations use the dynamic Smagorinsky model (Lilly, 1992) with the Lagrangian averaging of the model coefficient proposed by Meneveau et al. (1996). Each simulation is carried out for $100,000$ s, and statistics are horizontally and time averaged from $80,000$ to $100,000$ s.



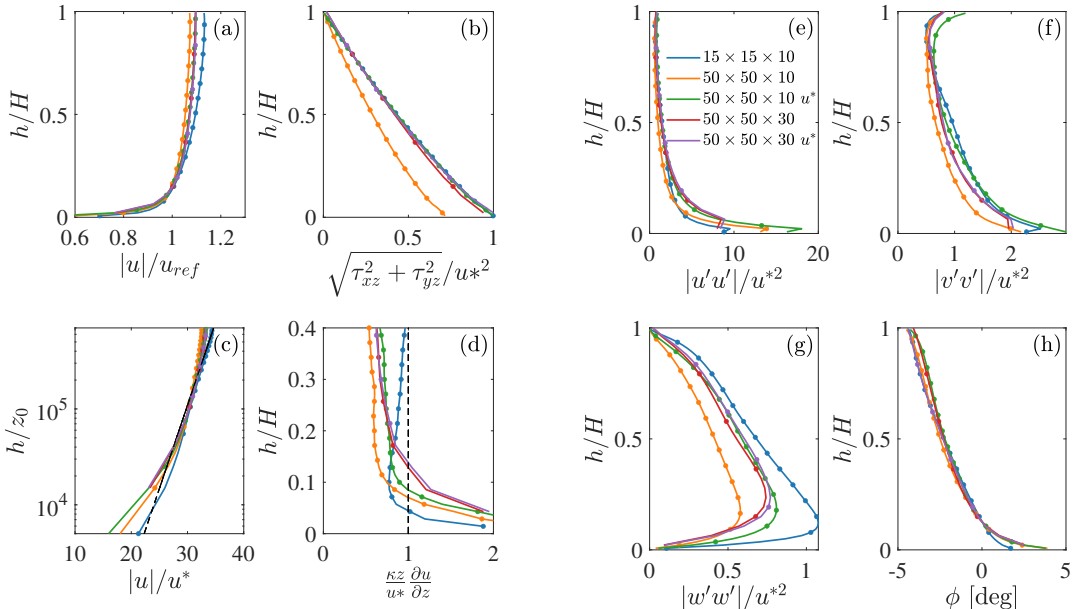

**Figure 3.** Statistics gathered from neutral ABL simulations having grid spacing of $15 \times 15 \times 10$ m, $50 \times 50 \times 10$ m and $50 \times 50 \times 30$ m. The modifier $u^*$ in the legend entries identifies those cases where the friction velocity in the wall model has been set equal to the $15 \times 15 \times 10$ m case, instead of being evaluated using Equation (17). All cases are normalized with $u_{\text{ref}} = 9$ m/s and $u^* = 0.297$ m/s (the latter corresponds to the case with mesh resolution of $15 \times 15 \times 10$ m); (a) and (b) show the velocity magnitude and shear stress profile, (c) and (d) depict the LOTW scaling and non-dimensional shear (theoretical laws are identified by the dashed black lines), (e), (f) and (g) report the mean velocity variances and (h) the wind veer resulting from the Coriolis force.

Figure 3 compares zero- and first-order statistics from the five precursor simulations. Firstly, it is evident how the simulation in general depends on the grid size when Brasseur and Wei (2010) criteria are not satisfied. The case depicting the largest deviation from the finest precursor corresponds to the $50 \times 50 \times 10$ m grid resolution without wall model correction. This case shows some difference in the mean velocity profile, produced by the LOTW mismatch and, more importantly, it completely fails to capture the vertical shear stress profile as well as strongly underestimates velocity variances. When the correct friction

velocity is applied to the wall model, things improve substantially. The shear stress profile closely matches the finer case and velocity variances also improve. The error in the mean velocity profile is reduced but a LOTW mismatch is still observed. When the vertical mesh resolution is decreased from 10 to 30 m, things improve when employing the conventional wall model and the shear stress almost matches with the fine precursor case. In fact, reducing the aspect ratio also reduces the LOTW mismatch according to Brasseur and Wei (2010), and so applying the correct friction velocity does not have a large impact on





the flow statistics. Surprisingly, the wind veer seems to show little sensitivity to the wall model and the grid size except when very close to the wall.

These results suggest that a precursor simulation characterized by a very large grid spacing — thus not designed to fulfill the LOTW matching criteria — can still capture the shear stress profile when the correct friction velocity is imposed at the wall. Moreover, a small deviation from the logarithmic profile, located at the boundary layer top, is observed when the velocity profile is compared to that of an LES satisfying LOTW scaling. Therefore, results from both the $50 \times 50 \times 30$ m case (with and without correction on $u^*$) and the $50 \times 50 \times 10$ m case (only with $u^*$ correction) are deemed suitable to provide an inflow condition to those wind farm simulations which, because they employ the AFM, are characterized by a comparably coarse grid.

## 2.5 Overset Mesh

The overset mesh technique (Benek et al., 1983) facilitates grid generation for flow around complex geometries as well as for bodies under relative motion (Meakin, 1993). It involves decomposition of the overall domain into a set of overlapping subdomains, such that the governing equations are independently solved in each of these domains and the information is transferred from one domain to the other at the subdomain interface using an interpolation method. Within wind energy applications, overset mesh method has been used for blade resolved simulations of wind turbines (Kirby et al.) as well as simulations capturing synoptic ($\approx 2000$ km), meso-scale ($\approx 100$ km), and micro-scale ($\approx 100$ m) effects simultaneously via nested static grids with both one-way and two-way coupling (Liu et al., 2011; Mirocha et al., 2013). TOSCA applies a one-way coupling strategy where the information from the background (coarser) grid is transferred to the enclosed overset (finer) grid at the overset mesh boundaries but no feedback is provided from the overset grid to the background grid. The AFM combined with an overset mesh method enables application of a finer mesh grid in a region of interest, such as a downstream wind farm in which the wind turbines are modeled using ADM.

Interpolation from the background mesh to the overset mesh begins with the identification of the face center points of the overset mesh cells along the interface, referred to as acceptor points. Then, for every acceptor point, the closest background mesh cell, referred to as donor cell, is identified. Using the relative position of the closest donor cell and the acceptor point, the eight donor cells from the background mesh that enclose an acceptor point are found and a tri-linear interpolation method is used to compute the velocity at the acceptor point. As the governing equations in TOSCA are formulated in generalized curvilinear coordinates, the numerical method solves for the contravariant fluxes instead of the Cartesian velocity (see Stipa et al., 2024a for further details), hence, from the interpolated velocity at the acceptor points, the boundary fluxes are calculated at each iteration to obtain the boundary information for the nested inner domain.

The trilinear interpolation scheme is non-conservative, hence, a correction of the local flux at the interface based on the mass residual is necessary to ensure global mass conservation. Here, the interpolated flux correction is made proportional to the flux similar to Zang and Street (1995) for global conservation of mass. A local flux proportional correction $\bar{U}^r$ for local flux $U^r$ is given as

$$\bar{U}^r = U^r - \frac{\epsilon_v |U^r|}{|U^r|_{\text{sum}}} \frac{\mathbf{n} \cdot \zeta_r}{|\mathbf{n} \cdot \zeta_r|}, \tag{20}$$





where $\epsilon_v$ is the global mass flux imbalance, $|U^r|_{\text{sum}}$ is the sum of the flux magnitude, $\mathbf{n}$ is the outward pointing unit normal to the overset boundary and $\zeta_r$ is the unit normal to the curvilinear co-ordinate line. Global mass flux imbalance $\epsilon_v$ summed over the interface cell faces is given by

$$\epsilon_v = \sum \left( U^r \frac{\mathbf{n} \cdot \zeta_r}{|\mathbf{n} \cdot \zeta_r|} \right). \tag{21}$$

With respect to the wind farm simulations presented in Sections 5 and 6, the above interpolation method is used for the overset domain at the streamwise inlet, spanwise lateral boundaries and the upper boundary. The streamwise outlet employs a zero normal gradient on velocity, while the wall model defined by Equations (17) to (19) is used at the bottom wall.

## 3 Isolated Wind Turbine

In order to confirm the selection of the AFM parameters, a parametric analysis is conducted by varying $r_{1/2}$, $s$, the velocity sampling strategy, and the horizontal mesh resolution for the uniform flow around an isolated wind turbine. Results are compared against two uniform ADM simulations characterized by a fine and coarse mesh resolution, respectively. The idealized simulations employed in this first phase offer crucial insights on the optimal choice of the AFM settings. However, it should be kept in mind that the AFM, as the name suggests, is developed to model wind farm clusters rather than isolated turbines. In fact, the knowledge obtained by conducting these idealized isolated wind turbine simulations will be later applied to the wind farm studies presented in Sections 4 to 6.

Regarding the two ADM simulations used for comparison, the coarser one employs a grid resolution of $30 \times 12.5 \times 10$ m in the streamwise, spanwise and vertical direction, respectively. In the finer ADM simulation, this initial mesh is gradually refined in all directions to reach a uniform resolution of 2.1 m around the wind turbine. This fine region where the mesh is uniform extends 1 diameter upstream of the turbine and 5 diameters downstream. In the vertical direction, it extends $\pm h_{\text{hub}}$ above and below the hub-height. Notably, the resolution of 12.5 m along $y$ in the coarse ADM case is motivated by the fact that this is close to the largest lateral cell size that allows to model the NREL 5MW wind turbine using the ADM, as it satisfies the requirement of 10 mesh cells along the rotor diameter, while the streamwise and vertical grid spacings are similar to previous wind farm ABL studies (see Stipa et al., 2024b; Wu and Porté-Agel, 2017, among others). Following Calaf et al. (2010), the ratio $\sigma/\max(\Delta y, \Delta z)$ in the ADM is set to 1.5 in order to avoid numerical oscillations when projecting the force from the actuator points. This leads to $\sigma/\Delta x = 0.625$, which is then extended to all AFM simulation since Equation (10) has a Gaussian shape in the streamwise direction. For the fine ADM case, where the mesh is uniform around the wind turbine, $\sigma$ is chosen such that $\sigma/\Delta = 1.5$. The projection error for the two ADM cases, evaluated as the relative difference between the cell-integrated body force after projection and the force sum from all actuator points is equal to 2.5% and 0.32% for the coarse and fine cases, respectively. Throughout the paper the wind turbine corresponds to the NREL 5MW reference turbine (Jonkman et al., 2009), characterized by a radius $R$ of 63 m and a hub height $h_{\text{hub}}$ of 90 m.

Regarding the choice of the AFM parameters, the ratio $r_{1/2}/R$ is set to 0.8, 1 and 1.2 while the smoothing $s$ is set to 2, 6 and 10. For each combination of these parameters, two types of velocity sampling methods are tested, namely the rotor disk and the





integral sampling methods. Finally, four different mesh resolutions are chosen in the spanwise direction, namely 12.5 m, 20 m, 40 m and 60 m. In the streamwise direction, previous studies showed evidence that an accurate solution can be obtained with a mesh resolution as large as 30 m, so the latter is used in conjunction with the values of 12.5 m and 20 m along $y$. For the 40 m and 60 m grids, mesh cells are rendered equal also in the streamwise direction. In the vertical direction, all cases feature a mesh resolution of 10 m, which corresponds to a representative value for wind farm ABL LESs. For all cases, periodic boundary

conditions are applied at the spanwise boundaries, while a slip condition is enforced at the upper and lower boundaries. At the outlet, a zero normal gradient on velocity outflow is specified, while the inlet is set to a uniform velocity of 9 m/s. The turbine rotor is 600 m away from all boundaries except from the lower one, which is located at a distance equal to $h_{\mathrm{hub}}$. This forces to use representative values of $r_{1/2}$ and $f$ to ensure that the projection function decays to zero before reaching the ground. As all four mesh configurations are uniform, they lead to the number of cells for each simulation reported in Tab. 2. In total, this

isolated turbine parametric study involves 72 simulations.

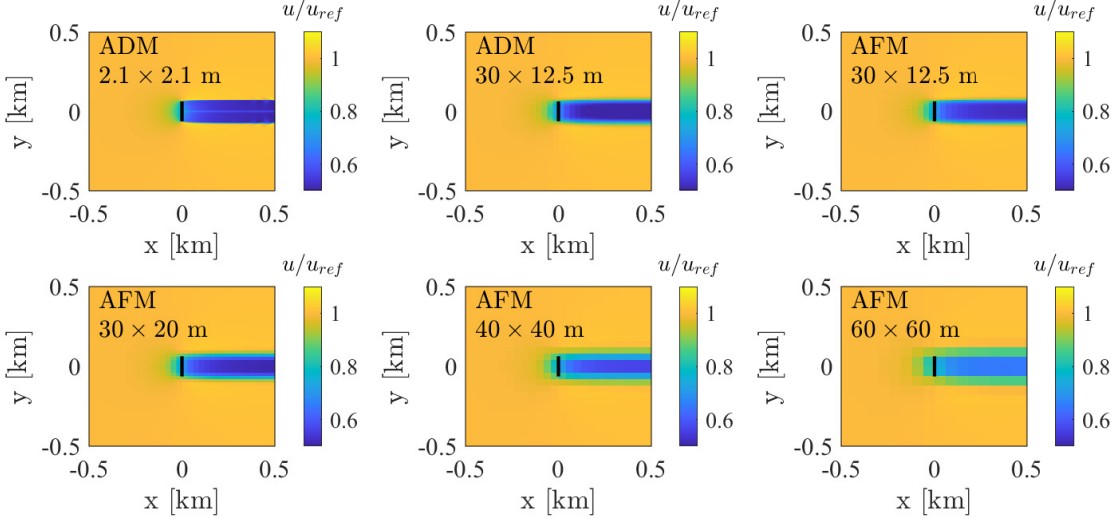

**Figure 4.** Velocity magnitude on an horizontal plane passing through the hub height for (top-left) ADM case with $2.1 \times 2.1 \times 2.1$ [m] grid resolution, (top-center) ADM case with $30 \times 12.5 \times 10$ [m] grid resolution, (top-right) AFM case with $30 \times 12.5 \times 10$ [m] grid resolution, (bottom-left) AFM case with $30 \times 20 \times 10$ [m] grid resolution, (bottom-center) AFM case with $40 \times 40 \times 10$ [m] grid resolution, (bottom-right) AFM case with $60 \times 60 \times 10$ [m] grid resolution.

The velocity field resulting from the two ADM cases and from four AFM cases characterized by the same AFM settings ($r_{1/2} = R$ and $s = 6$) and different mesh resolution is qualitatively shown in Figure 4. As can be noticed, the ADM and AFM models predict a very similar velocity field when the same mesh is employed. The ADM model predicts a slightly higher velocity deficit than the AFM model (see the remainder of this section for a quantitative comparison), which is due to a

spuriously increased turbine radius when accumulating the body force over all actuator points and to an increased body force towards the rotor center due to the body force being accumulated also from the neighbouring points. This does not occur for




the finer ADM case, where the standard deviation of the Gaussian projection function is reduced from $18.75$ m to $4.1$ m. In fact, body force accumulation from neighbouring points is minimal in this case and the actual wind turbine diameter is well represented. The coarsest AFM case is shown to visualize the flow field when the mesh is drastically coarsened. For this case, we also investigated the dependency of the AFM results to the relative position of the rotor disk with respect to the surrounding mesh cells (not shown here) and found very little sensitivity when the smoothing is greater than $6$.

| $\Delta x \times \Delta y \times \Delta z$ | $Nx \times Ny \times Nz$ | $N_{\text{dofs}}$ |
|---|---|---|
| 30-2.1 $\times$ 12.5-2.1 $\times$ 10-2.1 | $390 \times 260 \times 186$ | 18 860 400 |
| $30 \times 12.5 \times 10$ | $40 \times 96 \times 69$ | 264 960 |
| $30 \times 20 \times 10$ | $40 \times 60 \times 69$ | 165 600 |
| $40 \times 40 \times 10$ | $30 \times 30 \times 69$ | 62 100 |
| $60 \times 60 \times 10$ | $20 \times 20 \times 69$ | 27 600 |

**Table 2.** Number of mesh cells and total number of degrees of freedom for each mesh configuration of the isolated wind turbine cases. The first line of the table corresponds to the fine ADM case, where the mesh is graded in each direction to reach a resolution of 2.1 m around the wind turbine. In all other cases the cell size is constant in each direction.

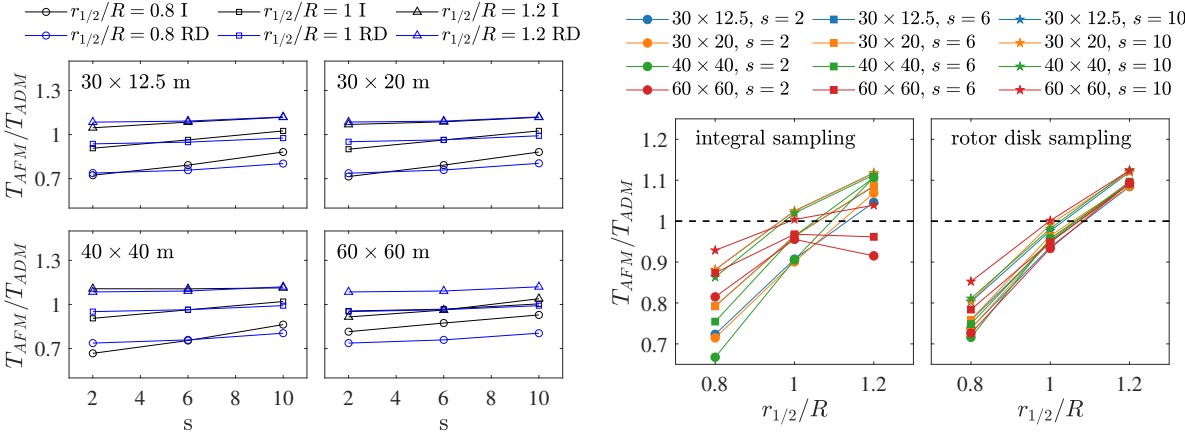

**Figure 5.** In the left four panels, AFM to ADM thrust ratio $T_{AFM}/T_{ADM}$ for all AFM cases performed in the present section. Blue and black colours indicate rotor disk (RD) and integral (I) sampling, respectively, while different symbols indicate different values of $r_{1/2}/R$. The flatness parameter is reported on the x-axis and data on the same panel are obtained using the same mesh resolution. The two right panels report AFM to ADM thrust ratio $T_{AFM}/T_{ADM}$ for all cases performed in the present section, where the left and right panels correspond to the integral and rotor disk sampling methods, respectively. The line colour identifies the mesh resolution, while each symbol corresponds to a different value of the flatness parameter. The $r_{1/2}/R$ ratio is shown on the x-axis.





Figure 5 shows the metric $T_{AFM}/T_{ADM}$, where $T_{ADM}$ and $T_{AFM}$ are the turbine thrust obtained with the ADM and AFM models, respectively, for all the cases conducted in this section. The ratio $T_{AFM}/T_{ADM}$ is not very sensitive to both the smoothing parameter and the mesh resolution, especially for the rotor disk sampling method. Conversely, the results seems to
be greatly affected by the $r_{1/2}/R$ ratio, with ratios lower and higher than unity underestimating and overestimating turbine thrust, respectively. This behaviour is confirmed by looking at the two rightmost panels, where each panel contains all cases characterized by the same sampling method. For each value of $r_{1/2}/R$, the integral sampling method is more sensitive to the smoothing parameter and mesh resolution than the rotor disk sampling, which is expected because the sampled velocity is directly related to the spatial support of the projection function. The rotor disk sampling shows very little spread for any given
value of $r_{1/2}/R$. For both sampling methods, $r_{1/2}/R = 1$ appears to provide the least error on wind turbine thrust.

Figure 6 shows the vertical profiles of velocity magnitude at different streamwise locations and at a spanwise coordinate coincident with the rotor center. The mesh resolution is characterized by different symbols, while the velocity sampling method is identified by their colour. All data correspond to $s = 6$ and each panel refers to a different value of $r_{1/2}/R$. As noticed previously, AFM results are very sensitive to the value of $r_{1/2}/R$. When $r_{1/2}/R$ is low the same turbine force has to be
distributed over a smaller volume, thus increasing the body force locally. Conversely, when $r_{1/2}/R$ is large, the body force decreases as the force is projected over a larger volume. As a result, setting $r_{1/2}/R = 1$ represents the best choice to capture the velocity field around the wind turbine for all the investigated values of grid spacing and velocity sampling methods.

In Figure 7, the same analysis is performed by fixing $r_{1/2}/R = 1$ and studying the dependence of the velocity profile on the smoothing parameter $s$. For $|y| < R$, varying the smoothness leads to a slight overestimation of the wake deficit for low $s$
(more so when using the rotor disk sampling method), and an underestimation for high values of $s$. The opposite behavior can be observed for $|y| > R$. This behavior is expected, as increasing $s$ increases the spatial support of the projection function, thus increasing the apparent radius of the wind turbine. Although the variation with $s$ is small, $s = 6$ seem to produce the best match in terms of velocity deficit when this is compared against ADM simulations, regardless of the mesh resolution. Moreover, some differences can be observed between the fine and coarse ADM simulations, where the smoothing generated by increasing the
standard deviation of the Gaussian projection function leads to a wake deficit overestimation due to an increased body force towards the rotor center and to a smearing of the velocity profile when transitioning from the wake to the outer flow.

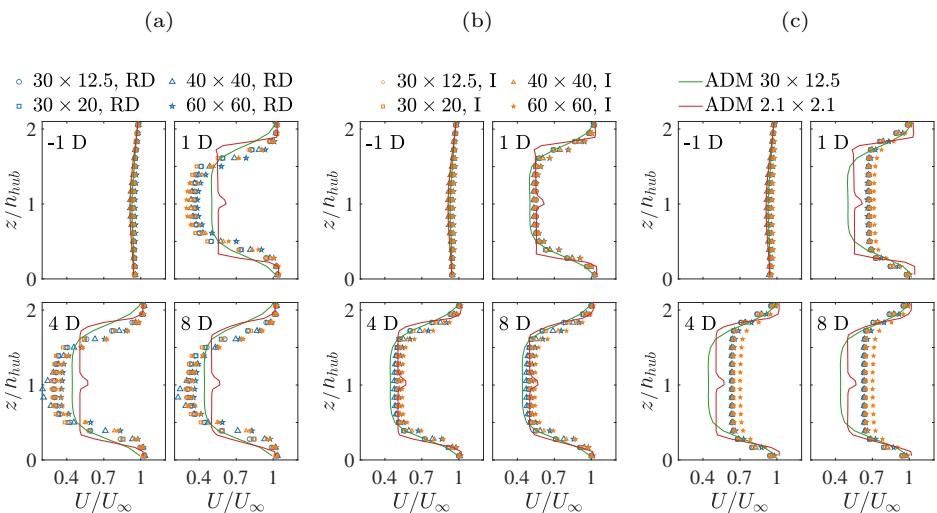

**Figure 6.** Vertical velocity profile at $y = 0$ for $s = 6$ and (a) $r_{1/2}/R = 0.8$, (b) $r_{1/2}/R = 1$, (c) $r_{1/2}/R = 1.2$. Symbols indicate different mesh resolutions, while colors refer to the velocity sampling strategy. Red and green lines indicate the fine and coarse ADM simulations, respectively. Each sub-panel corresponds to a different streamwise location, indicated in the figure.

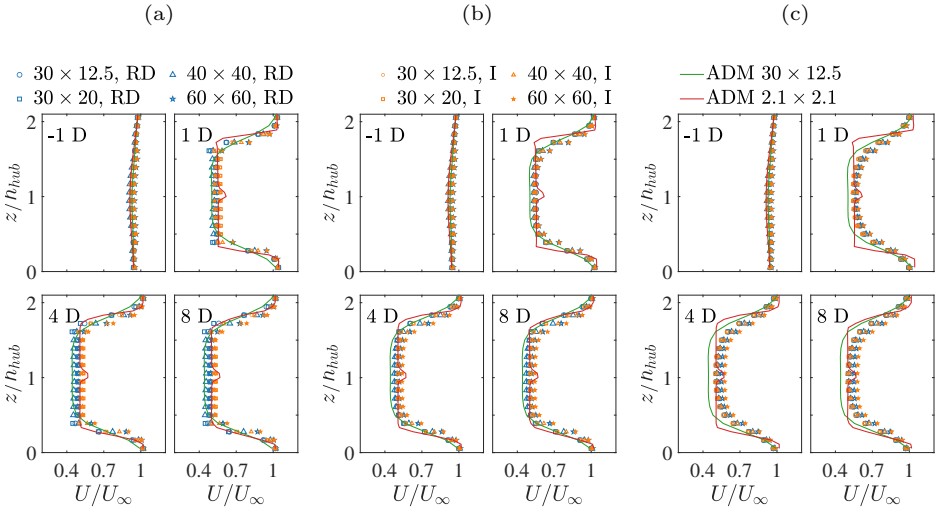

**Figure 7.** Vertical velocity profile at $y = 0$ for $r_{1/2}/R = 1$ and (a) $s = 2$, (b) $s = 6$, (c) $s = 10$. Symbols indicate different mesh resolutions, while colors refer to the velocity sampling strategy. Red and green lines indicate the fine and coarse ADM simulations, respectively. Each sub-panel corresponds to a different streamwise location, indicated in the figure.



## 4   Isolated Wind Farm

This section describes the wind farm simulations setup and results. Both an aligned and a staggered wind farm consisting of 25 wind turbines, organized in 5 rows and 5 columns, are investigated. When the wind turbines are modeled using the AFM, the four different mesh resolutions employed for the isolated wind turbine simulations described in Section 3 are used for each wind farm configuration. This allows to study the sensitivity of both the AFM and the LES to the grid resolution. For each wind farm, a baseline case employing the ADM is conducted, characterized by a mesh resolution of $30 \times 12.5 \times 10$ m, in the streamwise, spanwise and vertical directions, respectively, similar to previous numerical setups by Stipa et al. (2023a) and Lanzilao and Meyers (2023). In all cases, wind turbines are immersed in the same neutral ABL described in Section 2.3. In particular, while further advancing both the $15 \times 15 \times 10$ m and $50 \times 50 \times 10$ m cases in time for $20,000$ additional seconds, $y - z$ slices of velocity are saved at each time step in what is referred to as the inflow database. The coarser precursor employs the wall model correction described in Section 2.3, where the value of $u^*$ is obtained from the finer precursor. The generated inflow databases are used to prescribe the inlet velocity field for the wind farm simulations by linearly interpolating in time from the two closest available time samples, as well as bi-linearly interpolating in space from the precursor to the successor two-dimensional boundary meshes. In order to avoid the mismatch in the shear stress profile when the ratio between the target over source mesh size is greater than 2, the inflow data from the $15 \times 15 \times 10$ m precursor is only used for the successor cases characterized by a grid spacing of $30 \times 12.5 \times 10$ m and $30 \times 20 \times 10$ m, whereas the wind farm analyses involving a cell size of $40 \times 40 \times 10$ m and $60 \times 60 \times 10$ m use the inflow data obtained from the $50 \times 50 \times 10$ m precursor with wall-model correction. At the outlet, all wind farm simulations employ a zero gradient condition, while the remaining boundaries are treated similarly to their respective precursor simulation.

The wind farm is characterized by a streamwise spacing $S_x$ of 630 m (5 rotor diameters) and a spanwise spacing $S_y$ of 600 m (4.76 rotor diameters). For the aligned case, this yields a total size $L_x^f \times L_y^f$ of $2.4 \times 2.52$ km in the streamwise and spanwise directions, respectively, while the same values of $S_x$ and $S_y$ determine a total wind farm size of $2.4 \times 2.82$ km for the staggered case, as rows 2 and 4 are shifted by $-S_x/2$ in the spanwise direction. The successor domain is $15.6 \times 8.4 \times 0.7$ km, arranged such that 3 km are left on each side of the wind farm (for the staggered case they reduces to 2.7 km on the bottom side) and between the domain inlet and the first wind farm row. This leads to 10.08 km between the last wind farm row and the domain outlet, which are used to track the wind farm wake evolution. All four mesh configurations are uniform, leading to the number of cells for each simulation reported in Table 3. The AFM settings are based on the results from Section 3, hence $r_{1/2}/R = 1$ and $s = 6$. The streamwise standard deviation $\sigma$ is set to be consistent with the ADM simulations, where the isotropic standard deviation is set to $1.5\Delta_y = 18.75$ m. This corresponds to $\sigma = \Delta x/1.6$ for the finer AFM case, which is maintained for all mesh resolutions. In total, we run 8 AFM simulations and 2 ADM simulations. All analyses are advanced in time for $20,000$ s and flow statistics are averaged for the last $15,000$ s.




| $\Delta x \times \Delta y \times \Delta z$ | $Nx \times Ny \times Nz$ | $N_{\text{dofs}}$ |
|---|---|---|
| $30 \times 12.5 \times 10$ | $520 \times 672 \times 70$ | 24 460 800 |
| $30 \times 20 \times 10$ | $520 \times 420 \times 70$ | 15 288 000 |
| $40 \times 40 \times 10$ | $390 \times 210 \times 70$ | 5 733 000 |
| $60 \times 60 \times 10$ | $260 \times 140 \times 70$ | 2 548 000 |

**Table 3.** Number of mesh cells and total number of degrees of freedom for each mesh configuration of the wind farm cases.

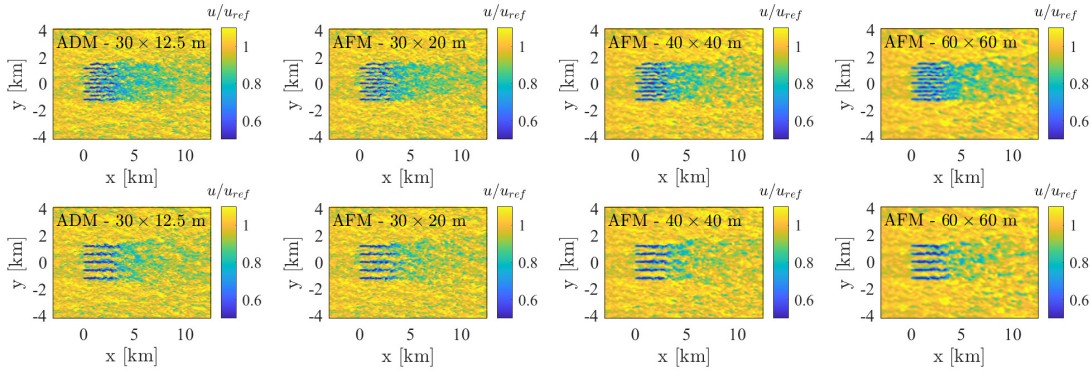

**Figure 8.** Contours of instantaneous hub height velocity field from the ADM and AFM simulations. Top and bottom panels correspond to the aligned and staggered wind farm layouts, respectively.

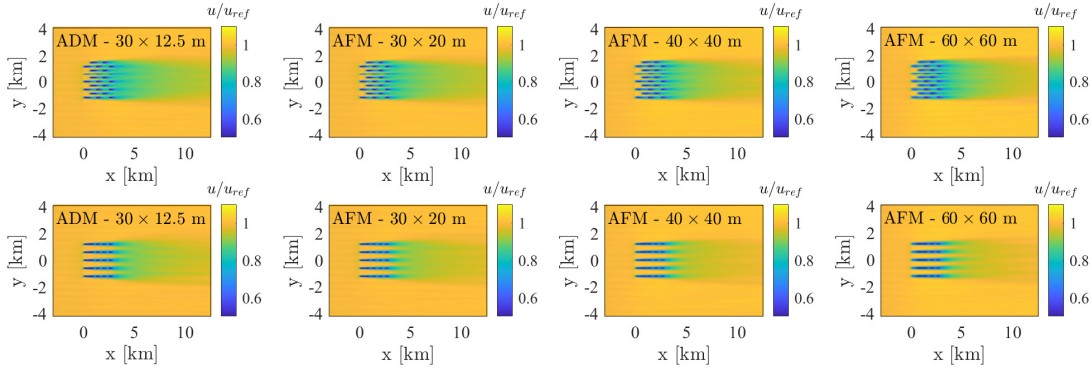

**Figure 9.** Contours of time-averaged hub height velocity field from the ADM and AFM simulations. Top and bottom panels correspond to the aligned and staggered wind farm layouts, respectively.

In Figure 8, the contours of instantaneous velocity field at the hub height are reported for all simulations apart from the $30 \times 12.5 \times 10$ m AFM cases. Individual wind turbine wakes show a lower tendency to meander for the AFM simulations characterized by a lower horizontal grid resolution ($40 \times 40$ m and $60 \times 60$ m). Although this may result in a slower individual




wake recovery than in the higher mesh resolution cases, it is expected, as reducing the grid resolution increasingly filters both the incoming ABL turbulence and the fine flow features that characterize each individual turbine wake. However, as can be noticed by looking at averaged hub height velocity field reported in Figure 9, once turbine wakes have merged, the wake of the entire cluster is less sensitive on the grid's ability to capture the evolution of each individual turbine wake, and cases
characterized by different turbine model and mesh resolution are in very good agreement.

From the row-averaged thrust and power reported for all cases in Figure 10, it can be noticed that the ADM and the AFM yield very similar results for the $30 \times 12.5 \times 10$ m resolution, for both the aligned and staggered cases. At the waked rows, the AFM model predicts a slightly lower values of thrust and power. This can be attributed to the employed velocity sampling strategy, according to which the wind speed is sampled at a single location. Notably, when an upstream aligned turbine is
present, the sampling location coincides with the wake centerline, leading to a lower sampled velocity. This aspect is mitigated for the staggered case, where aligned turbines are separated by a greater distance. At the non-waked rows, AFM predictions are fairly independent on the grid spacing. Conversely, at the waked rows, the AFM predicts a lower thrust and power as the grid resolution is reduced. The reason for such underestimation follows from the lower tendency for individual turbine wakes to meander when the grid size is increased.

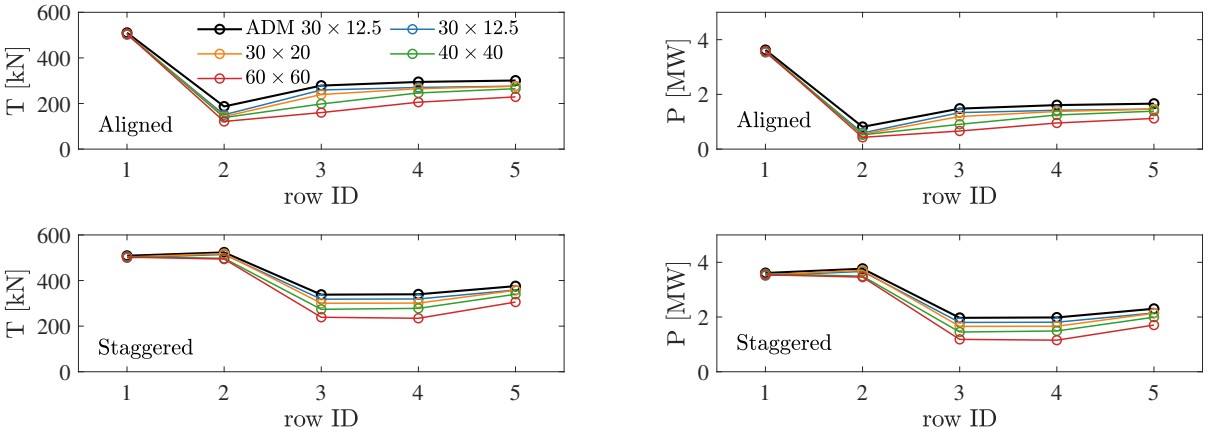

**Figure 10.** Row and time averaged thrust (left) and power (right) distributions obtained for the aligned (top) and staggered (bottom) wind farm layouts. The black line refers to the ADM case, while the blue, orange, green and red lines correspond to the results obtained using the AFM with rotor disk sampling, on the $30 \times 12.5$ m, $30 \times 20$ m, $40 \times 40$ m and $60 \times 60$ m horizontal mesh sizes.

Figure 11 shows the time-averaged spanwise velocity profiles at the hub height at different streamwise locations inside the wind farm and in the wake, for both the aligned and staggered wind farm layouts. As can be observed, the spanwise velocity profiles predicted using the AFM are in good agreement with the ADM results, both inside and downstream of the wind farm, for both the staggered and the aligned layouts.





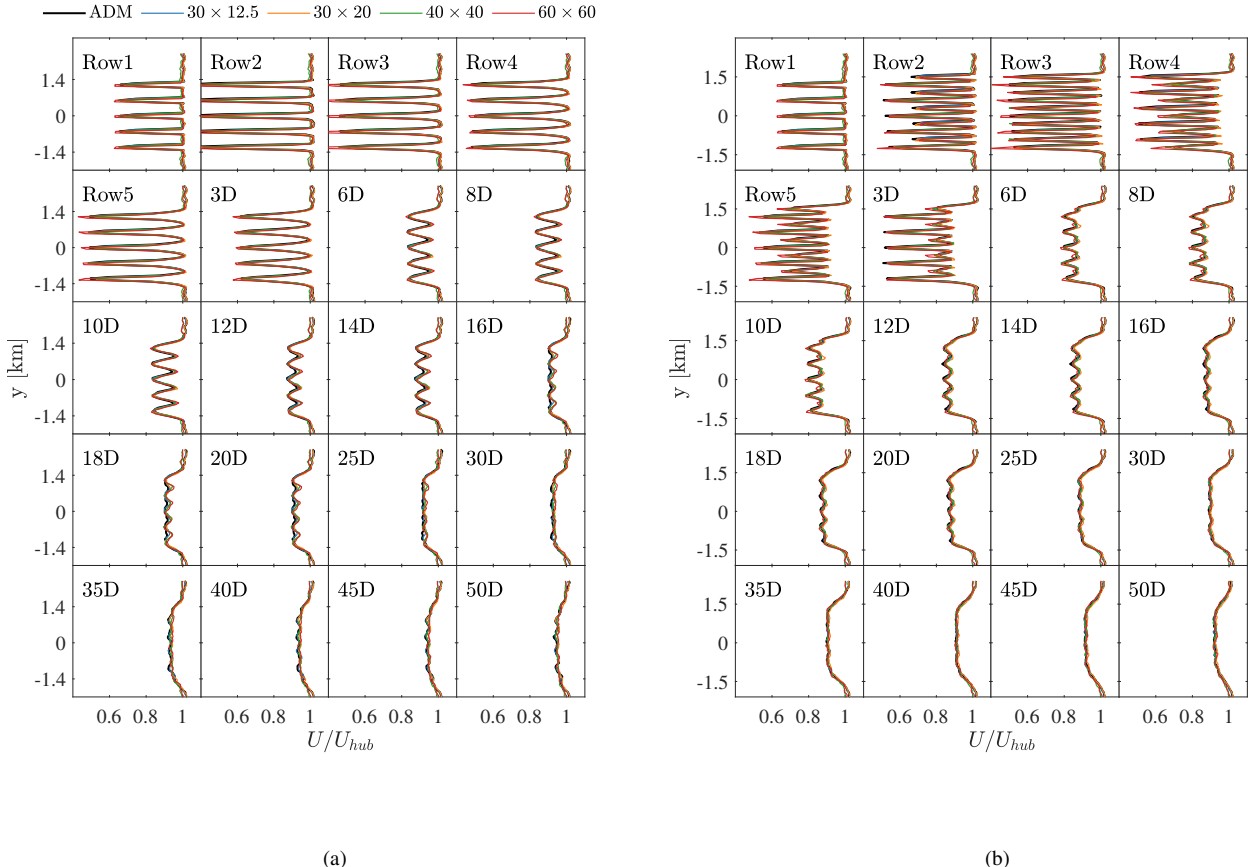

(a)                                                                              (b)

**Figure 11.** Time averaged spanwise velocity profiles at the hub height, sampled at different streamwise locations inside the wind farm and in the wake, for the aligned (left) and staggered (right) layouts. Wind farm locations are identified with the row ID, while wake locations are identified by their distance in rotor diameters from the last wind farm row. The black line refers to the ADM case, while the blue, orange, green and red lines correspond to the results obtained using the AFM with rotor disk sampling, on the $30 \times 12.5$ m, $30 \times 20$ m, $40 \times 40$ m and $60 \times 60$ m horizontal mesh sizes.

Figure 12 reports the time-averaged hub height velocity as a function of the streamwise coordinate, further averaged over
the wind farm width. As can be noticed, except from the AFM results obtained with the $60 \times 60$ m horizontal mesh resolution, the velocity evolution agrees well with that predicted by using the ADM model, especially upstream of the wind farm and in the wake. In fact, it can be argued that the wind speed in the wind farm wake is fairly independent of the grid spacing after the individual turbine wakes have merged.

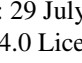



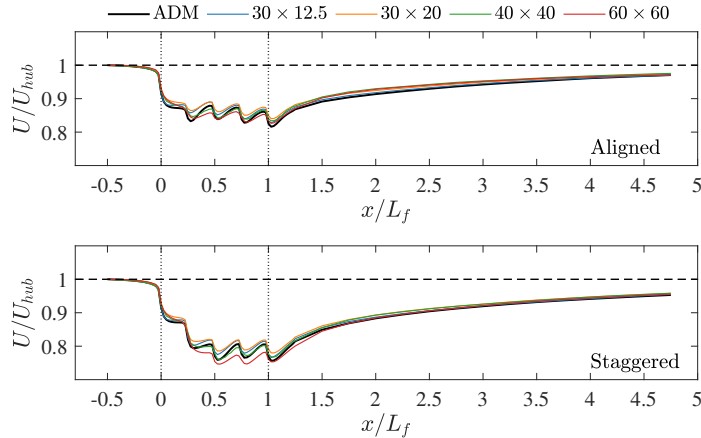

**Figure 12.** Streamwise evolution of time-averaged hub height velocity, further averaged over $y = \pm 2.5$ km, for the aligned (top) and staggered (bottom) layouts. The black line refers to the ADM case, while the blue, orange, green and red lines correspond to the results obtained using the AFM with $30 \times 12.5$ m, $30 \times 20$ m, $40 \times 40$ m and $60 \times 60$ m horizontal mesh sizes. Vertical dashed lines correspond to first and last wind farm rows.

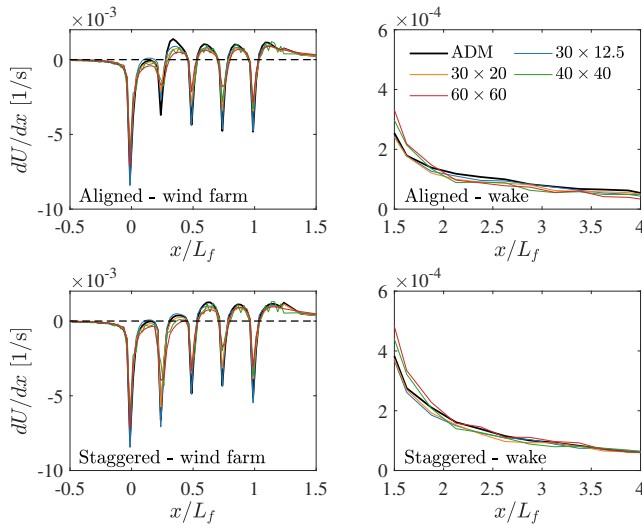

**Figure 13.** Streamwise derivative of the time-averaged hub height velocity, further averaged over the wind farm width, for the aligned (top) and staggered (bottom) layouts. The black line refers to the $30 \times 12.5 \times 10$ m ADM case, while the blue, orange, green and red lines correspond to the results obtained using the AFM with rotor disk sampling, on the $30 \times 12.5$ m, $30 \times 20$ m, $40 \times 40$ m and $60 \times 60$ m horizontal mesh sizes. Left panels refer to the wind farm region, right panels refer to the wake region.





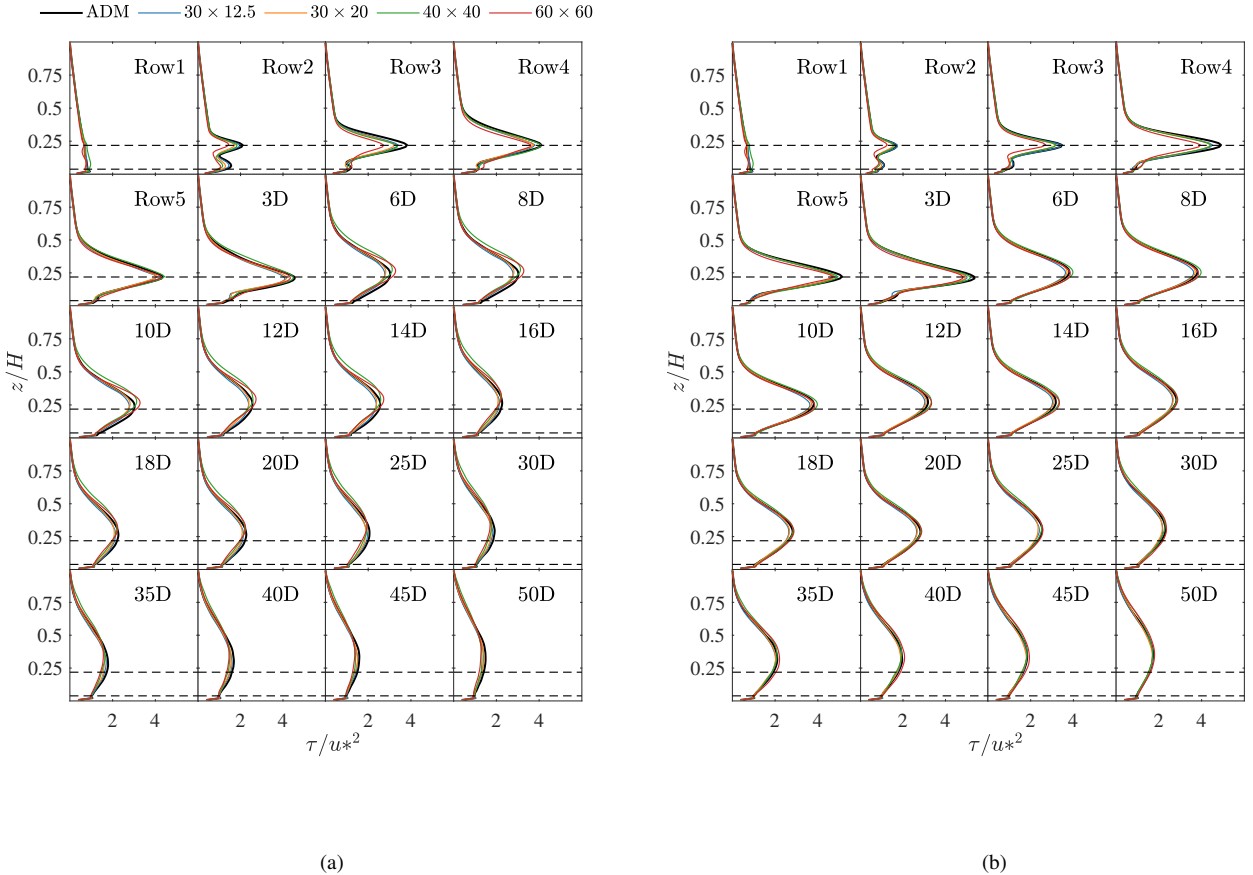

(a)

(b)

**Figure 14.** Time-averaged vertical shear stress profiles, further averaged over the wind farm width, for the aligned (left) and staggered (right) layouts. Wind farm locations are identified with the row ID, while wake locations are identified by their distance in rotor diameters from the last wind farm row. The black line refers to the ADM case, while the blue, orange, green and red lines correspond to the results obtained using the AFM with rotor disk sampling, on the $30 \times 12.5$ m, $30 \times 20$ m, $40 \times 40$ m and $60 \times 60$ m horizontal mesh sizes. Horizontal dashed lines refer to $h_{\text{hub}} \pm R$.

To further expand on this and assess the differences in wake recovery predicted by the AFM using different grid sizes, we
report in Figure 13 the streamwise derivative of the mean velocity previously shown in Figure 12. As can be noticed, while the $40 \times 40 \times 10$ m and the $60 \times 60 \times 10$ m mesh resolutions under-predict wake recovery inside the wind farm with respect to the finer meshes, wake recovery is well captured by all mesh resolution after $\approx 2$ wind farm lengths downstream of the last turbine row, with the largest deviations observed for the $60 \times 60$ AFM case with the aligned wind farm layout.

One of the effects of the wind farm on the ABL flow is to increase vertical turbulent mixing by enhancing the level of shear
stress. This enhances momentum entrainment from above the wind farm, playing an important role in the wake recovery of the entire wind farm. Figure 14 shows the time-averaged vertical shear stress profiles, further averaged over the wind farm width, at different streamwise locations inside the wind farm and in the wake. Notably, the ADM and AFM results are generally



in good agreement except for the AFM case characterized by the largest grid spacing, which under predicts the shear stress profile evolution inside the wind farm. In the wind farm wake, all cases are in very good agreement, for both the aligned and

staggered layouts. The same conclusions can be drawn from the time-averaged vertical velocity profile at $y = 0$ m, reported in Figure 15. In addition, it is evident from this figure how the first cell velocity strongly depends on the employed grid spacing when the LOTW scaling is not captured according to Brasseur and Wei (2010) criteria. This applies to the $40 \times 40 \times 10$ m and the $60 \times 60 \times 10$ m grids, where the velocity at the first cell decreases as the horizontal grid size increases. However, as discussed in Section 2.3 and confirmed here, this does not impair the results of the wind farm simulations, especially when the

wall shear stress experienced at infinity matches that of a simulation that complies with the LOTW scaling criteria. Moreover, even the coarsest grid seems to capture the shear stress perturbation generated by the wind farm, which explains why wind farm wake recovery is also well captured by all values of mesh resolution.

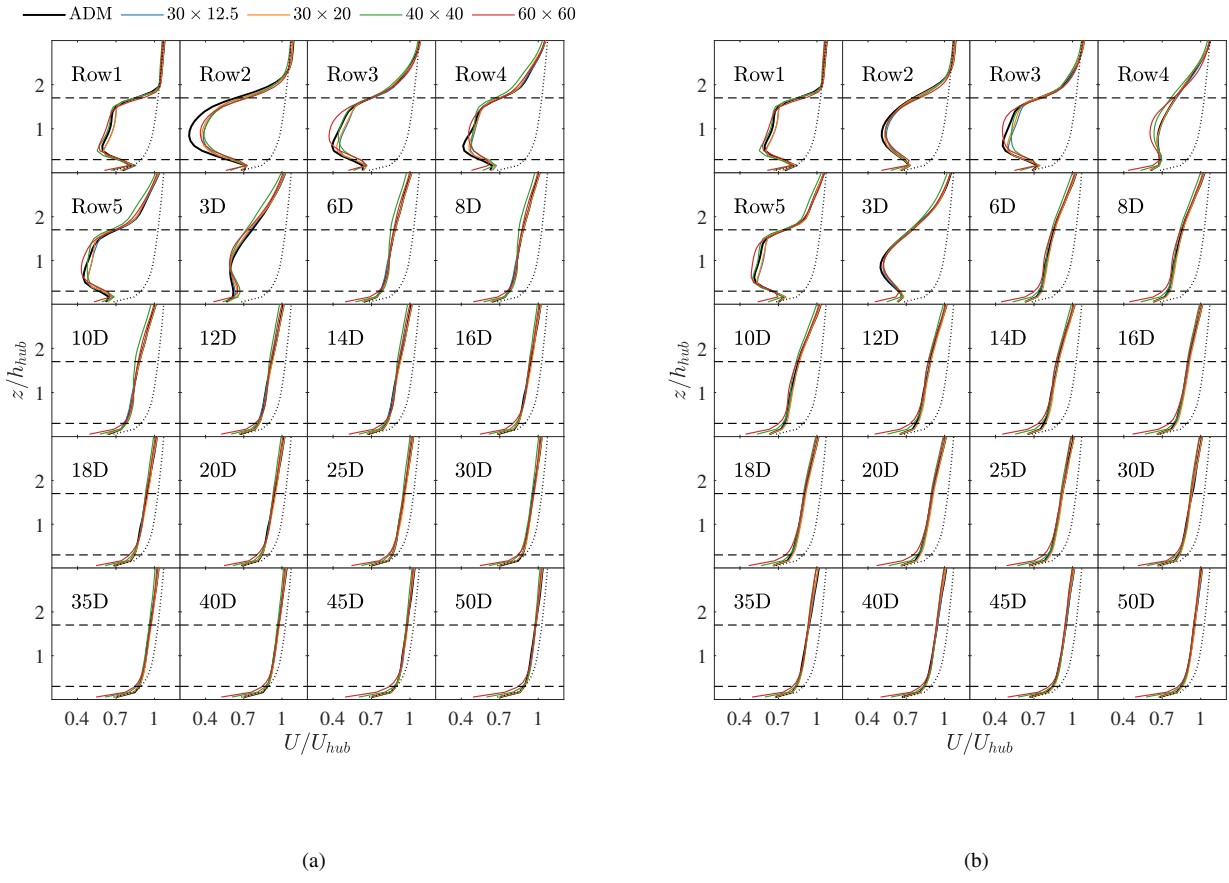

(a)             (b)

**Figure 15.** Time-averaged vertical velocity profiles at $y = 0$ m for the aligned (left) and staggered (right) layouts. Wind farm locations are identified with the row ID, while wake locations are identified by their distance in rotor diameters from the last wind farm row. The black line refers to the ADM case, while the blue, orange, green and red lines correspond to the results obtained using the AFM with rotor disk sampling, on the $30 \times 12.5$ m, $30 \times 20$ m, $40 \times 40$ m and $60 \times 60$ m horizontal mesh sizes. Horizontal dashed lines refer to $h_{\mathrm{hub}} \pm R$.





In summary, regarding the accuracy of the AFM with respect to the ADM, the two approaches are practically equivalent if the same mesh resolution is employed. For the $40 \times 40 \times 10$ m and the $60 \times 60 \times 10$ m grid sizes, the AFM captures the wind

farm power at the non-waked rows, while power is under-predicted at the waked turbines. We argue that this is not an issue of the AFM, but it is rather attributable to the inability to properly capture individual wake meandering by these coarser grids, leading to a slower recovery of individual turbine wakes. Nevertheless, all values of grid resolution can accurately capture the velocity distribution both upstream and downstream of the wind farm. Inside the wind farm, velocity profiles agree reasonably with those predicted by the ADM except for the AFM case characterized by a resolution of $60 \times 60 \times 10$ m, which over predicts

the wake deficit. As a consequence, a mesh characterized by a horizontal resolution of $40 - 60$ m may be employed for those problems where an array of interest is waked by an upwind wind farm, to discretize the flow region upstream of the array of interest. Conversely, the wake of the target turbine array should be discretized with no more than a 30 m horizontal resolution in order to properly capture individual wake interactions. This value of grid resolution corresponds to 4 cells along the rotor diameters for the wind turbine employed in the present study. Notably, this is still too coarse to use the ADM, highlighting the

cost-saving potential of the AFM.

## 5    Farm-Farm Interaction

In this section, we conduct simulations of two interacting wind farm clusters with the objective of understanding if the AFM can be used to model the upstream wind farm cluster at a low computational cost, when the main focus is on the downstream wind farm. We choose an idealized case where two aligned wind farms corresponding to the staggered layout of Section 4 are

separated by a distance of 5 km. The domain extends for $23 \times 8.4 \times 0.7$ km in the streamwise, spanwise and vertical direction, respectively. In a first case, all turbines are modeled using the ADM and the mesh resolution is set to $30 \times 12.5 \times 10$ m. All boundaries are treated similar to the isolated wind farm simulations described in Section 4. The inflow data corresponds to the fully neutral ABL described in Section 2.3 where the grid spacing is set to $15 \times 15 \times 10$ m. A second simulation employs one-way coupled nested domains using the technique described in Section 2.5. The size of the outer domain coincides with

that of the ADM case, but it is discretized using a $50 \times 50 \times 10$ m mesh resolution instead. Moreover, both wind farms are modeled using the AFM. The inner domain, characterized by a grid resolution of $30 \times 12.5 \times 10$ m, extends for $16 \times 8 \times 0.7$ m and its inlet boundary is located 4 km after the start of the first wind farm. Here, wind turbines are modeled using the ADM and velocity is interpolated at the inlet, top as well as the side boundaries from the outer domain. At the wall, a wall model based on the classic Monin and Obukhov (1954) similarity theory is employed while the outlet is treated similar to the outer

domain.

Since we employ a one-way domain nesting, the downwind wind farm is modeled in both the outer and inner domains to capture its effect on the upstream cluster. In order to highlight the effects of mapping the inflow database between different precursor and successor grid sizes, previously described in Section 2.3, the simulation employing the AFM is carried out twice, both using the same inflow database as the ADM simulation, as well as the inflow database obtained employing a precursor

mesh resolution of $50 \times 50 \times 10$ m with wall model correction. These two inflow databases correspond to those used in Section 4.





The boundary conditions in the outer domain are the same as the ADM case except for the wall. Specifically, when the inflow database generated from the coarse precursor is used, the wall shear stress is applied by fixing the friction velocity as described in Section 2.3, calculated from the finer precursor. Conversely, when the inflow database generated from the finer precursor is used to prescribe the inflow to the outer domain, classic Monin and Obukhov (1954) similarity theory is used to apply the wall shear stress. Table 4 summarizes the main features of the three simulations, such as the employed turbine model, the total number of mesh cells and the inflow data used to prescribe the ABL flow at the inlet. Throughout the remainder of this section, simulation 1,2 and 3 in Table 4 will be referred to as ADM, AFM and AFM with coarse inflow. All simulations are carried out for $20,000$ s and flow statistics are gathered for the last $15,000$ s.

| Case | | Farm A | Farm B | Precursor | $\Delta x \times \Delta y \times \Delta z$ | $Nx \times Ny \times Nz$ | $N_{\text{dofs}}$ |
|---|---|---|---|---|---|---|---|
| 1 | | ADM | ADM | $15 \times 15 \times 10$ | $30 \times 12.5 \times 10$ | $766 \times 672 \times 70$ | $36\,032\,640$ |
| 2 | outer | AFM | AFM | $15 \times 15 \times 10$ | $50 \times 50 \times 10$ | $460 \times 168 \times 70$ | $29\,288\,000$ |
| | inner | n/a | ADM | | $30 \times 12.5 \times 10$ | $533 \times 640 \times 70$ | |
| 3 | outer | AFM | AFM | $50 \times 50 \times 10$ + | $50 \times 50 \times 10$ | $460 \times 168 \times 70$ | $29\,288\,000$ |
| | inner | n/a | ADM | wall model corr. | $30 \times 12.5 \times 10$ | $533 \times 640 \times 70$ | |

**Table 4.** Summary of turbine model, mesh size used in the precursor and successor simulations and number of degrees of freedom for the wind farm simulations conducted in the present section.

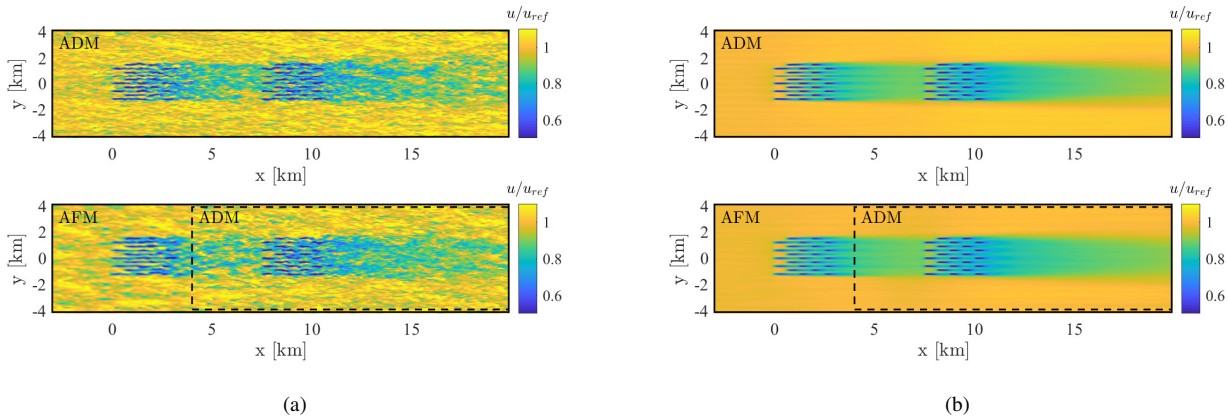

(a)                  (b)

**Figure 16.** Instantaneous (left) and time-averaged (right) hub height wind speed obtained from cases 1 and 3 of Table 4. The dashed black line shows the horizontal size of the inner domain in the AFM simulation.

Figure 16 shows the contours of instantaneous and time-averaged hub height velocity for the AFM with coarse inflow and ADM cases. Although the coarser mesh used in the outer domain of the AFM case filters out the fine turbulence structures that are instead resolved in the ADM case, a very good agreement exists between the two on a qualitative level. A more quantitative




comparison between all cases is reported in Figure 17 by showing the mean vertical velocity profile at $y = 0$ and the mean shear stress profiles, further averaged over $y = \pm 2.5$ km.

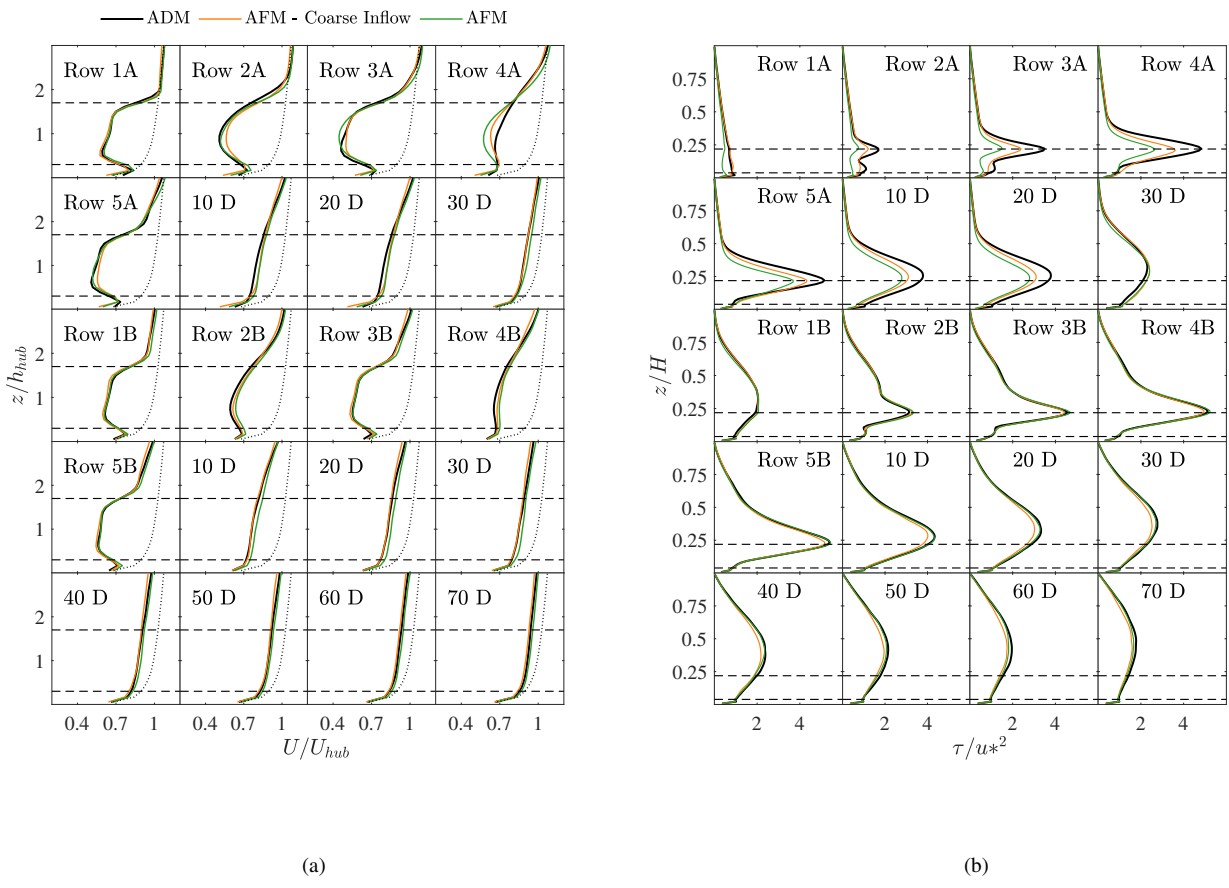

**Figure 17.** Left: time-averaged vertical profiles of velocity magnitude at $y = 0$ and at different streamwise locations. Right: time-averaged vertical shear stress profile, further averaged between $y = \pm 2.5$ km, at different streamwise locations. In both panels, streamwise locations are identified by the row number and wind farm ID (A for upstream and B for downstream), when inside a wind farm, or by their distance in rotor diameters from the last row of the closest upstream wind farm. The dashed line in the left panel corresponds to the freestream velocity obtained from the precursor simulations characterized by the $15 \times 15 \times 10$ m grid resolution. Horizontal dashed lines refer to $h_{\text{hub}} \pm R$.

First, the vertical velocity profiles predicted by the ADM and AFM simulation that employs the coarse inflow agree well at every streamwise location. Only a slightly higher velocity is observed up to 20 rotor diameters downstream of the first wind farm. Conversely, the AFM simulation that uses the finer inflow consistently underestimates the wake deficit by the same amount in both wind farm wakes. Looking at the shear stress profile, the largest difference can be observed around the first wind farm, while all profiles collapse after 30 rotor diameters downstream of the first cluster. In particular, it can be noticed how the shear stress profile is strongly under predicted at first wind farm row by the AFM case where the finer inflow database is used.





This issue, previously described in Section 2.3, reaches all the way to the inlet of the outer domain and it is attributed to the velocity mapping from the inflow database. Specifically, when this is interpolated from the finer precursor mesh to the coarser successor grid, non-solenoidal velocity fluctuations are produced which are subsequently altered by the pressure iteration of our solver when it corrects the velocity field. Conversely, when the coarser inflow data and the wall model correction are used, the inlet shear stress profile agrees with that resulting from the ADM case. Moreover, both AFM cases underestimate — to an

acceptable extent when the coarser inflow is used — the perturbation in shear stress inside the first wind farm. Notably, this is an expected mechanism when the mesh size is increased, since also the LES filter size grows and more eddies are modeled by an increase in the eddy viscosity. Further downstream, as the flow enters the inner domain, the shear stress profile from the different cases are in good agreement with each other.

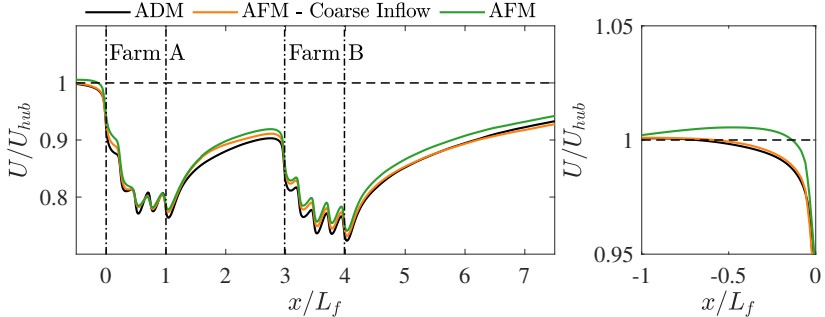

**Figure 18.** Streamwise evolution of time-averaged hub height wind speed, further averaged between $y = \pm 2.5$ km from the three cases conducted in the present sections. Vertical dashed lines indicate the two wind farms. A magnification of the left panel in the induction region of the first wind farm is depicted in the right panel.

The effect on velocity of the reduction in shear stress at the domain inlet operated by the mapping procedure when the source

and target grids are very different from each other can be clearly visualized from Figure 18. In fact, looking at the blockage region of the AFM case employing the inflow database generated with the finer grid, one can see that the wind speed increases after the inlet, before reaching the wind farm. This, phenomenon, explained in Section 2.3, is due to the fact that a reduction in shear stress causes a deformation of the velocity profile in order to satisfy the momentum budget inside the boundary layer. This may potentially induce also a spanwise velocity components, as the momentum source terms representing the constant driving

pressure gradient in the successor simulation are averaged from the precursor simulation, thus they require the same shear stress profile to fulfill the horizontal momentum balance. The most important consequence of such issue is that the blockage region is completely misrepresented and the freestream velocity experienced by the upstream wind farm is increased. However, by applying the inflow data calculated using the coarse mesh and the correction to the wall model, the shear stress profile is not altered and the velocity in the induction region is correctly captured. In general, using this approach results in good agreement

with results from the ADM case. The largest differences are observed in the wake of the first wind farm and are likely due to the wind farm thrust under prediction by the AFM when the horizontal resolution exceeds $\approx 30$ m. To some extent, also the





fact that the AFM in general predicts a slightly lower freestream wind speed — and thus a lower thrust — due to the employed sampling method may play a minor role.

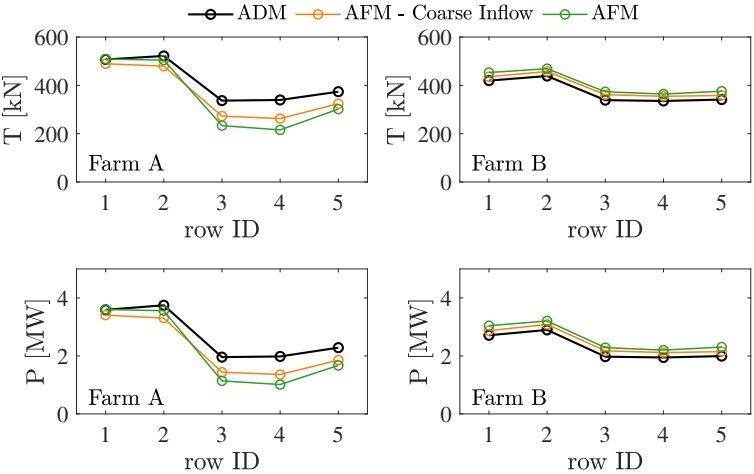

**Figure 19.** Row-averaged thrust (top) and power (bottom) distributions from the three cases described in this section, for the upstream (left) and downwind wind farm (right).

Finally, Figure 19 shows the row-averaged wind turbine thrust and power for the two wind farms obtained from the three
different cases. For the two AFM cases, data correspond to the AFM and ADM models for the upwind and downwind wind farms, respectively. In general, the AFM results point out to the same conclusions as the isolated wind farm cases presented in Section 4, i.e. that a horizontal resolution of $50 \times 50$ m is not sufficient to capture the absolute thrust and power. However, trends are reasonably captured and the error reduces when the coarser inflow database is used, as the shear stress profile — and thus the turbulence intensity level — agree with the ADM case. In addition, thrust and power from the downwind wind farm are
only slightly overestimated, again with the error decreasing if the coarse inflow data and wall model correction are used. These results suggest that the AFM is a good candidate model for problems involving one or more wind farms waking a downstream wind farm of interest. In particular, the less stringent requirement on mesh resolution imposed by the AFM reduces the overall computational cost while still capturing the cluster wake evolution with reasonable accuracy.

## 6   Wind Farm-Induced Atmospheric Gravity Waves

In this section, we conduct LES of wind farm-induced atmospheric gravity waves (AGW), investigating the ability of the AFM to capture the AGW evolution in the free atmosphere. In particular, we simulate a wind farm immersed in a conventionally neutral boundary layer (CNBL), i.e. a boundary layer developing against a potential temperature stratification characterized by a neutral region, followed by a capping inversion layer with strength $\Delta\theta$ and thickness $\Delta h$, centered at $H$, and a linear stable lapse rate $\gamma$ aloft. For the CNBL, we chose $\Delta\theta = 5$ K, $\Delta h = 100$ m, $\gamma = 4$ K/km and $H = 500$ m, which also coincides



with the height of the boundary layer, as its growth is limited by the capping inversion. These values are frequently observed offshore in the North Sea, as pointed out by Lanzilao and Meyers (2023). The equivalent roughness height $z_0$ is set to $0.0001$ m and the reference potential temperature $\theta_{\text{ref}}$ is equal to $300$ K. The precursor simulation uses a velocity controller which aims at maintaining a reference velocity of 9 m/s at the wind turbine hub height of $90$ m, as turbines correspond to the NREL 5MW reference turbine (Jonkman et al., 2009). The precursor simulation is advanced for $100,000$ s in order to spin up turbulence, and the horizontally averaged potential temperature profile is kept constant by the temperature controller described in Stieren et al. (2021). The flow is initialized with a uniform velocity of 9 m/s, while temperature follows the model developed by Rampanelli and Zardi (2004). Geostrophic damping using the same settings employed by Stipa et al. (2024b) is applied to remove inertial oscillations that may arise when the initial geostrophic speed is not in geostrophic balance, a condition that cannot be avoided when forcing the wind speed at a height located inside the boundary layer. The precursor domain extends for $6 \times 6 \times 1$ km in the streamwise, spanwise and vertical directions, respectively, and it is discretized using a grid size of $15 \times 15$ m in the horizontal directions. Below the start of the inversion layer, the vertical grid size is set to 10 m. From $450$ to $500$ m the grid is reduced to 5 m and then increased again to 10 m at $550$ m, to capture the Ellison scale within the inversion layer (Allaerts and Meyers, 2017). The 10 m resolution is then maintained until the upper boundary. After the first $100,000$ s of simulation, statistics are averaged for $20,000$ s and $y-z$ flow sections are saved at each time step to form the inflow database. The profiles of wind speed and direction, shear stress, and potential temperature from the precursor phase are reported in Figure 20, while quantitative data are summarized in Table 5.

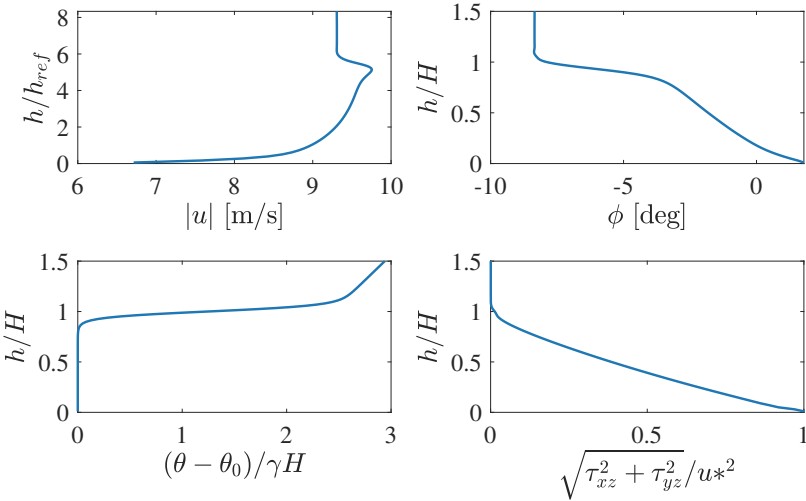

**Figure 20.** Vertical profiles of wind speed magnitude (top left), wind veer (top right), potential temperature (bottom left), and shear stress (bottom right) for the CNBL precursor. Data are averaged from 100.000 to 120.000 s.



| $G$ [m/s] | $u^*$ [m/s] | $q_{min} \cdot 10^4$ [Km/s] | $\phi_G$ [deg] |
|---|---|---|---|
| 9.31 | 0.26 | -0.63 | -8.35 |

**Table 5.** Values of the geostrophic wind, friction velocity, minimum heat flux $q_{min}$ within the boundary layer, and geostrophic wind angle obtained from the CNBL precursor.

Regarding the wind farm simulations, their setup it sketched in Figure 21. For the ADM case, this consists of a domain that extends for $22.62 \times 12$ km in the streamwise and spanwise directions, respectively. The wind farm corresponds to the staggered layout described in Section 4, where the first row is located at $x = 0$ and separated by 10 km from the inlet boundary.

Moreover, the wind turbines at the first row sides are located at a distance of 4.8 km from the lateral boundaries. Regarding the vertical domain size, this is dictated by the simulation's ability to resolve AGWs. In fact, the total domain height should be at least twice as the expected gravity wavelength $\lambda_z$ (this parameter can be estimated as $\lambda_z = 2\pi G/N$, where $N$ is the Brunt-Väisälä frequency and $G$ is the geostrophic wind). Moreover, a Rayleigh damping layer should be used at the upper boundary to avoid AGW reflection, characterized by a layer depth $\geq \lambda_z$ (Lanzilao and Meyers, 2022a). With reference to the

CNBL parameters used in the present study, $\lambda_z \approx 5.2$ km. Hence, the domain height has been set to 14 km, while the start of the Rayleigh damping region has been placed at 7 km (blue box in Figure 21). The mesh resolution in the vertical direction follows the precursor simulations below 1 km, while it is stretched up to 200 m in the Rayleigh damping region. Specific details are provided in Table 6. In order to also avoid AGW reflections from the inlet boundary, a fringe and an advection damping regions characterized by the same activation functions adopted by Lanzilao and Meyers (2022a) are used (magenta and orange boxes

in Figure 21, respectively), and their parameters are reported in Table 7. Following the same authors, the Rayleigh and fringe region damping coefficients have been set to $\nu_{RDL} = 0.035 \ s^{-1}$ and $\nu_{FR} = 0.03 \ s^{-1}$, respectively. The successor simulations employs lateral periodic boundary conditions, a slip wall at the upper boundary and a wall model based on the classic Monin and Obukhov (1954) similarity theory at the wall.

| $z_s$ [km] | $z_e$ [km] | $\Delta z$ [m] | $N$ [-] |
|---|---|---|---|
| 0 | 0.4 | 10 | 40 |
| 0.4 | 0.5 | 10-4.85 | 14 |
| 0.5 | 0.6 | 4.59-10 | 15 |
| 0.6 | 1 | 10 | 40 |
| 1 | 7 | 10-200 | 95 |
| 7 | 14 | 200 | 36 |

**Table 6.** Vertical discretization for successor and concurrent precursor simulations. The parameter $N$ indicates the number of cells in each mesh layer. These extend from $z_s$ to $z_e$ and are characterized by a cell size $\Delta z$.



As spatially- and time-resolved velocity and temperature fields that are unperturbed by the wind farm are required in the
fringe region to compute the damping source terms, a concurrent precursor simulation characterized by a domain size coincident with the fringe region ($3 \times 6 \times 14$ km) and having the same mesh resolution as the wind farm domain is carried on in
sync with the latter. The concurrent precursor uses inflow slices saved from the CNBL simulation described above to enforce a
time-resolved inflow condition. At the outlet we apply a zero gradient boundary condition, while all remaining boundaries are
treated similarly to the wind farm simulation. Since the flow slices available from the pre-computed inflow database are $6 \times 1$
km large in the spanwise and vertical directions, their data is tiled two times along $y$ and extrapolated along $z$ in order to be
mapped at the concurrent precursor inlet.

| $x_s$ [km] | $x_e$ [km] | $\Delta_s$ [km] | $\Delta_e$ [km] |
|---|---|---|---|
| $-10$ | $-7$ | $0.75$ | $0.75$ |

(a) Fringe region parameters.

| $x_s$ [km] | $x_e$ [km] | $\Delta_s$ [km] | $\Delta_e$ [km] |
|---|---|---|---|
| $-9$ | $-5$ | $1$ | $1$ |

(b) Advection damping region parameters.

**Table 7.** Fringe and advection damping region information.

| Case | | Turbine Model | Precursor | $\Delta x \times \Delta y \times \Delta z$ | $Nx \times Ny \times Nz$ | $N_{\text{dofs}}$ |
|---|---|---|---|---|---|---|
| 1 | | ADM | $15 \times 15 \times 10$ | $30 \times 12.5 \times 10/5/200$ | $754 \times 960 \times 240$ | $173\,721\,600$ |
| 2 | outer | AFM | $15 \times 15 \times 10$ | $50 \times 50 \times 10/5/200$ | $452 \times 240 \times 240$ | $50\,496\,000$ |
| | inner | ADM | | $30 \times 12.5 \times 10$ | $520 \times 672 \times 70$ | |

**Table 8.** Summary of turbine model, mesh size used in the precursor and successor simulations and number of degrees of freedom for the
wind farm simulations conducted in the present section.

Regarding the two AFM simulations, they employ two one-way nested domains. The outer domain setup coincides with
the ADM simulation described above, with the only differences being that the AFM is used in place of the ADM and that the
horizontal mesh resolution is coarsened to $50 \times 50$ m. The inner domain (black box in the left panel of Figure 21) extends for
$15.6 \times 8.4 \times 0.7$ km, with the inlet boundary placed at $x = -3$ km, and it is discretized using a mesh resolution of $30 \times 12.5 \times 10$
m. Velocity and potential temperature are interpolated at the lateral, upper and inlet boundaries from the outer domain, while
the outlet and bottom boundaries use a zero gradient and a wall model based on classic Monin and Obukhov (1954) similarity
theory, respectively. In the inner domain, wind turbines are modeled using the ADM. This arrangement allows AGWs to be
captured in the outer domain, where they are forced by the AFM, and their effects to be transferred to the inner domain by
interpolating velocity and potential temperature from the outer grid. Notably, in the outer domain, the same inflow data used
for the ADM simulation is used to provide an inlet boundary condition to the concurrent precursor. This means that data is
mapped from a $y - z$ grid resolution of $15 \times 10$ m, employed for the CNBL simulation, to a mesh spacing of $50 \times 10$ m. The
issue described in Section 2.3 related to the modification of the shear stress profile by the mapping interpolation is mitigated
by imposing the value of $u^*$ reported in Table 5 for both the concurrent precursor and outer domains. A summary is given





in Table 8 regarding the turbine model, precursor and successor grid size (that of the concurrent precursor coincides with the
successor) and number of degrees of freedom employed for both the ADM and AFM simulations. Both the ADM and AFM
cases are advanced in time for $20,000$ s and statistics are gathered during the last $15,000$ s of simulation.

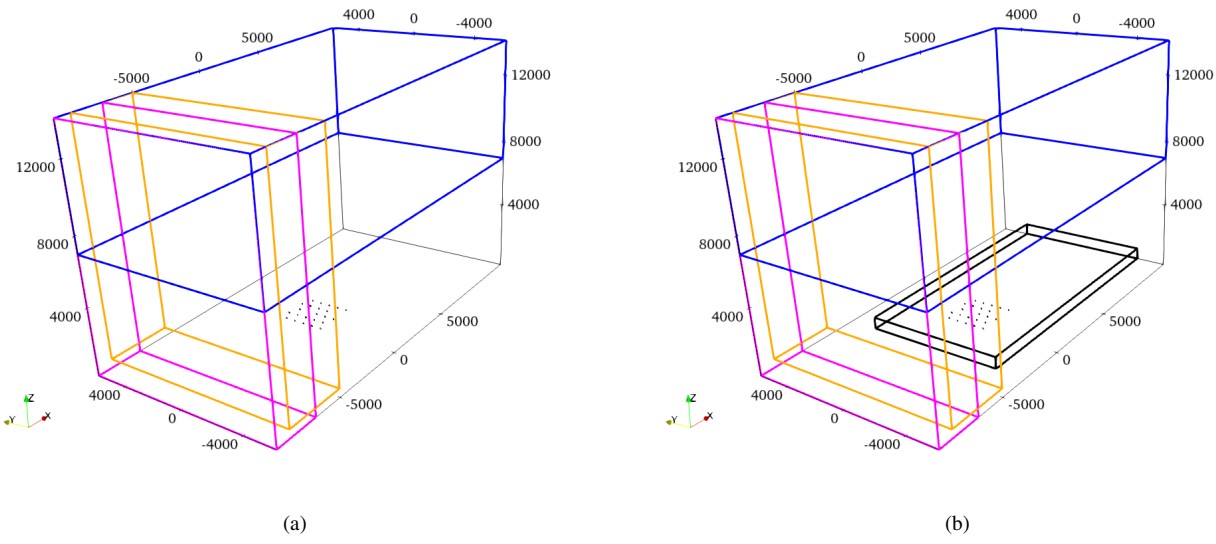

(a)                                                                          (b)

**Figure 21.** Location of the Rayleigh damping layer (blue), fringe region (magenta) and advection damping region (orange) relative to the
domain. The ADM case (left) only uses a single domain, while the AFM case (right) employs the same outer domain, characterized by a
coarser grid resolution and an inner domain (black) featuring the same grid spacing to that of the ADM case. Wind turbines are modeled
using the AFM in the outer domain while ADM is employed in the inner domain.

Figure 22 shows contours of time-averaged velocity at the hub height and the pressure perturbations produced at the same
height by the internal and interface waves triggered by the wind farm in the free atmosphere and within the inversion layer,
respectively. As can be noticed, the developed setup combining grid nesting with the AFM agrees well with the ADM results
employing a more conventional design of the numerical simulation. Differences are only observed inside the fringe region, i.e.
in a non-physical portion of the domain, where pressure perturbations are reduced for the AFM case. The reasons for such
difference are presently unknown to the authors, but it seems that the fringe region performs better when employing a coarser
grid resolution. Regarding the AGW patterns produced in the vertical velocity field and in the pressure perturbation field, they
can be visualized in Figure 23 on a $x - z$ plane located at $y = 0$. Also in this case, they agree extremely well between the
two simulations. For instance, the perturbations in pressure resulting from the AFM case seem to be slightly lower than that
predicted using the ADM but, as will be shown in the following analysis, they do not lead to a visible alteration of the results,
both in terms of wind speed and turbine quantities, when these are compared against the data extracted from the ADM case.





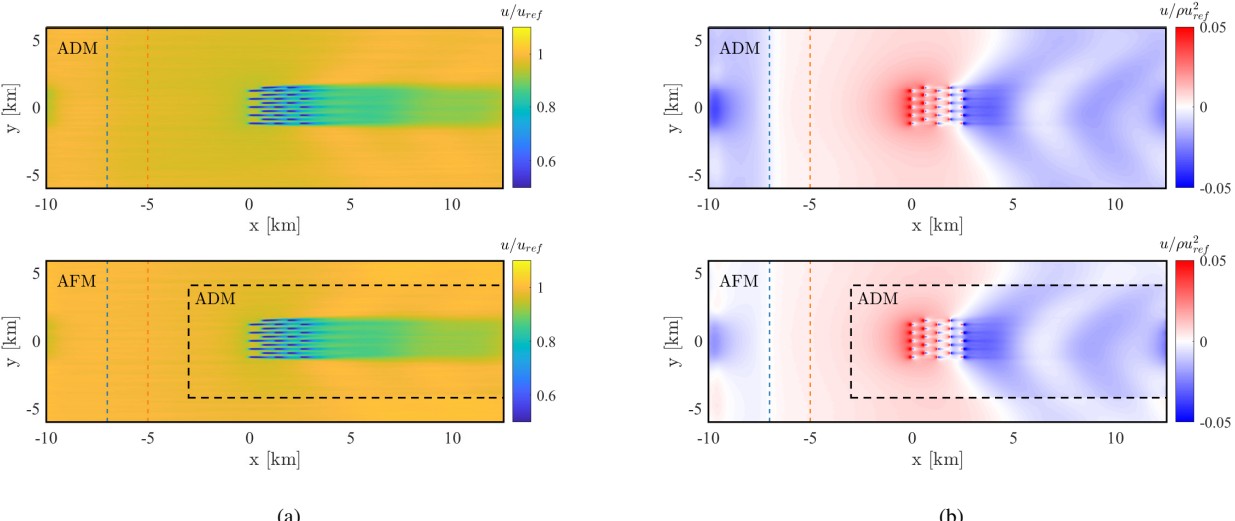

(a)                                                                    (b)

**Figure 22.** Time-averaged hub height velocity (left) and pressure perturbation (right), calculated as $p_p(x,y,z) = p(x,y,z) - p(x_\infty,y,z)$, where $x_\infty = -7$ km and $p(x,y,z)$ is the time-averaged pressure. Top and bottom panels correspond to the ADM and AFM cases, respectively. In the latter, the horizontal inner domain size is identified by the black dashed line. The end of the fringe and advection damping regions are identified by the dashed blue and orange lines, respectively.

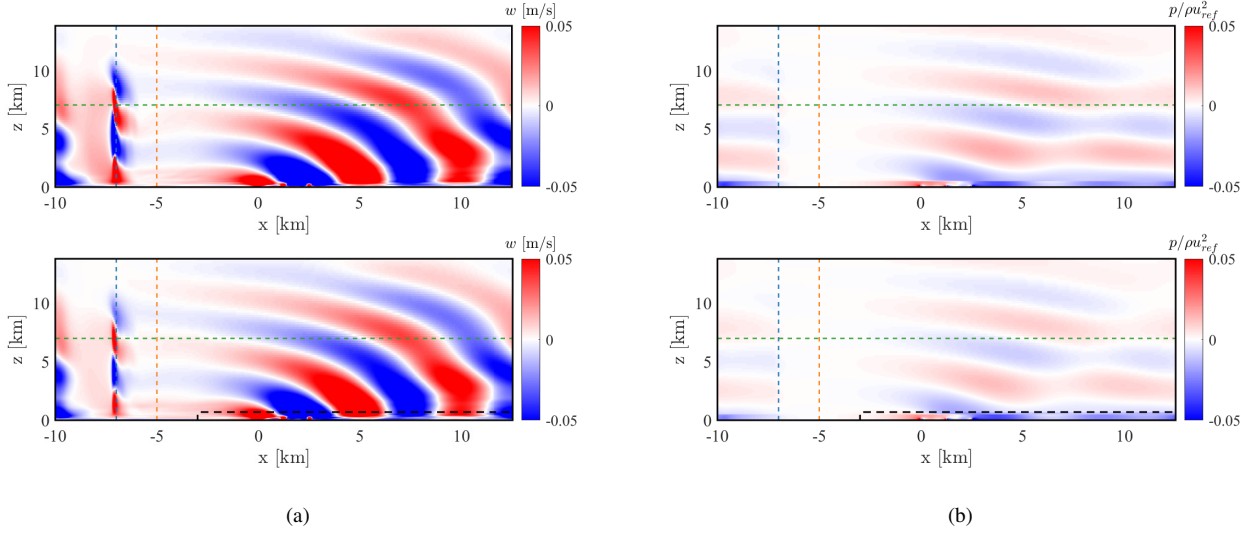

(a)                                                                    (b)

**Figure 23.** Time-averaged vertical velocity (left) and pressure perturbation (right) at $y = 0$. Top and bottom panels correspond to the ADM and AFM cases, respectively. In the latter, the horizontal inner domain size is identified by the black dashed line. The end of the fringe and advection damping regions are identified by the dashed blue and orange lines, respectively, while the start of the Rayleigh damping region corresponds to the green dashed line.



In this regard, Figure 24 reports the vertical flow perturbation, magnified 10 times, at different heights. The blue lines
correspond to the inversion layer displacement and dashed lines have been obtained using data from the outer domain in the
AFM simulation. As can be noticed, the perturbations obtained in the free atmosphere using the AFM are almost identical to
those obtained when wind turbines are modeled with the ADM, demonstrating that AGW are not sensitive to how accurately
the simulation captures the turbulent flow inside the boundary layer. These AGWs-induced vertical flow perturbations are
transferred to the inner domain when interpolating the velocity and temperature fields at its boundaries in the AFM simulation,
allowing to model AGWs effect on the wind farm and boundary layer flow.

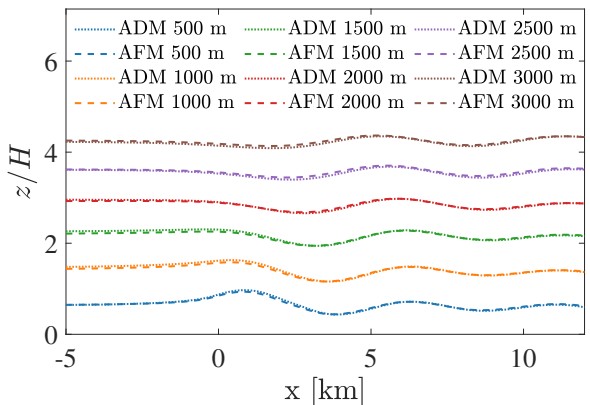

**Figure 24.** Vertical flow perturbation at different heights for the ADM (continuous line) and AFM (dashed line) magnified 10 times.

Figure 25 plots the vertical velocity profiles at $y = 0$ and the vertical shear stress profiles, averaged over $y = \pm 2.5$ km, at
different streamwise location inside and downstream of the wind farm. Data from the AFM simulation are entirely contained
within the inner domain, where wind turbines are modeled using the ADM. For instance, the coarse resolution in the outer
domain in the AFM simulation does not alter the incoming boundary layer flow when data are compared against the ADM
case. Only small differences in shear stress and velocity exist at the first wind farm row, but these are soon removed by the
higher grid resolution adopted in the inner grid and are not propagated downstream. In general, the two simulations are in
very good agreement, despite the AFM case involving 30% of the number of degrees of freedom as the ADM case. The effect
of AGWs in the free atmosphere on the boundary layer flow can be appreciated in Figure 26, where the time-averaged and
hub-height velocity and pressure fields, further averaged over the wind farm width are displayed. First, the pressure oscillations
in the wind farm wake induced by the lee waves previously shown in Figure 22 induce oscillations in velocity, leading to an
intermittent recovery of the wind farm wake. Moreover, if compared to previous results obtained with the same wind farm
in Section 4 and a boundary layer height of 0.7 km, wind farm blockage is greatly increased for the atmospheric conditions
analyzed in this section, and important reductions in velocity can be observed up to $\approx 0.5$ km upstream of the first wind farm
row. Note that, according to Smith (2023), a fully neutral boundary layer where the domain height coincides with the ABL
height corresponds to the rigid-lid approximation of very high stratification above the boundary layer. In particular, as shown



in Stipa et al. (2024a), allowing the inversion layer to displace according to the AGW solution in the free atmosphere modifies the pressure field around the wind farm, yielding different values of blockage and individual wake recovery inside the wind farm. Finally, Figure 27 reports the row-averaged turbine power and thrust from the two simulations, showing an excellent agreement between the two.

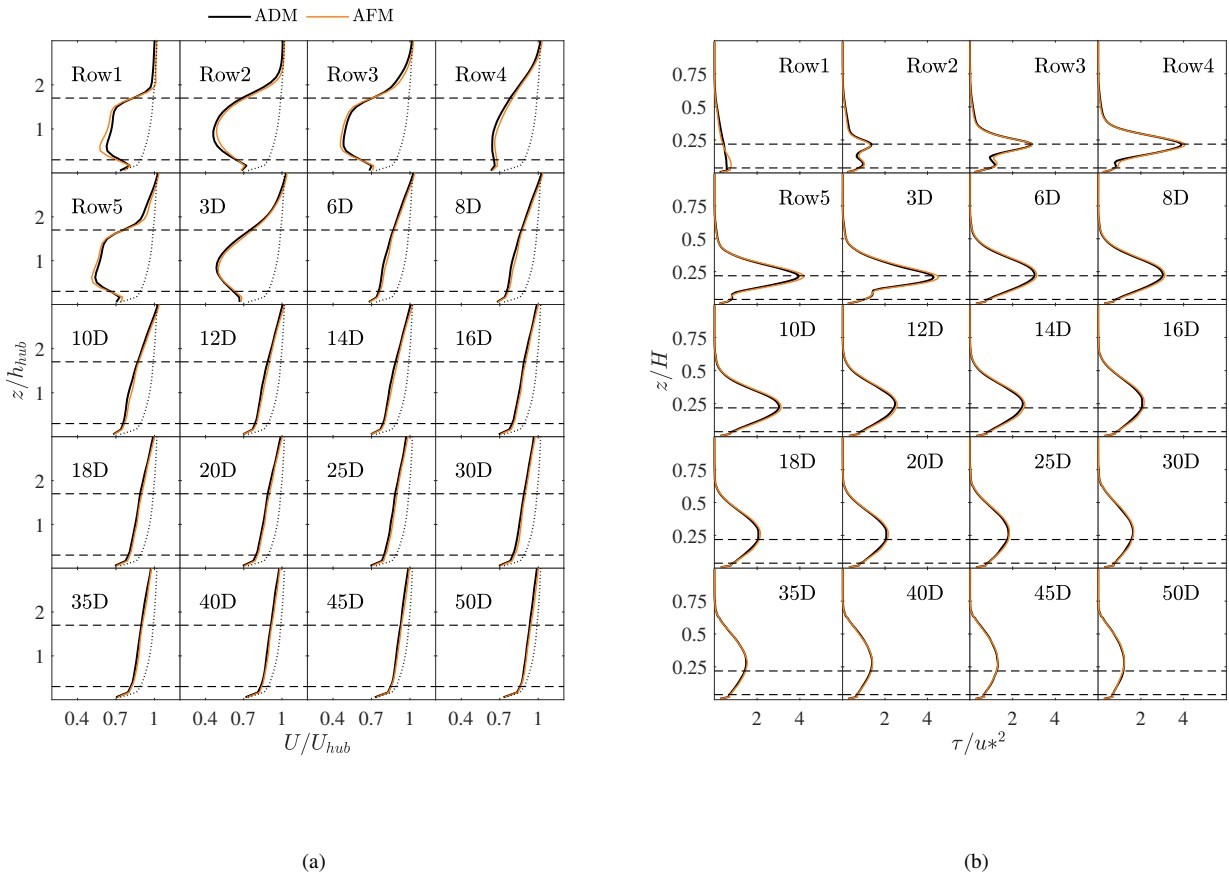

(a)  (b)

**Figure 25.** Left: time-averaged vertical profiles of velocity magnitude at $y = 0$ and at different streamwise locations. Right: time-averaged vertical shear stress profile, further averaged between $y = \pm 2.5$ km, at different streamwise locations. In both panels, streamwise locations are identified by the row number when inside a wind farm, or by their distance in rotor diameters from the last wind farm row. The dashed line in the left panel corresponds to the freestream velocity obtained from the precursor simulation. Horizontal dashed lines refer to $h_{\text{hub}} \pm R$.

Overall, these results demonstrate that, if our focus is only in the atmospheric flow solution, the AFM alone (i.e. without need of an inner domain) is sufficient to accurately capture AGWs in the free atmosphere and their effects on the ABL flow at a reduced computational cost. However, if accurate wind turbine information is required, an inner domain characterized by a higher grid resolution can be placed around the wind farm, and the outer flow — which contains the AGW solution — can be interpolated at the boundaries to model AGW effects on the wind farm performance. Notably, this is similar to the





model proposed by Stipa et al. (2024a) to account for AGW effects on the wind farm flow without extending the domain into the free atmosphere. However, while in Stipa et al. (2024a) the top boundary coincides with the inversion layer and it is physically displaced in order to enforce a slip boundary condition, here the slip boundary condition — which corresponds to no penetration — is replaced by interpolating the velocity from the outer domain, thus allowing some degree of permeability to the top boundary.

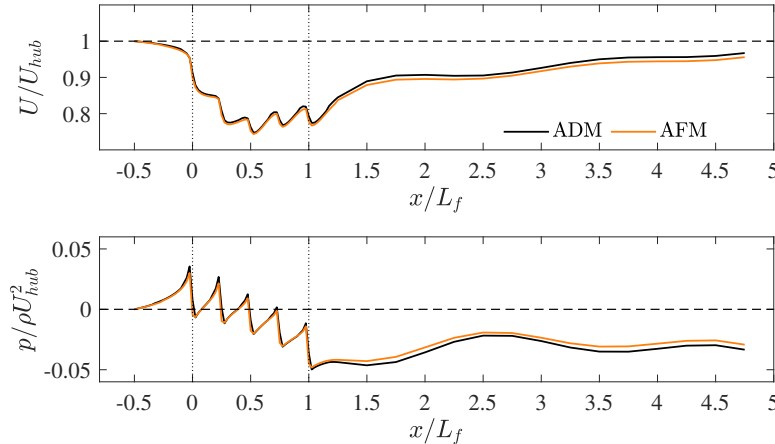

**Figure 26.** Streamwise evolution of time-averaged hub height velocity (top) and hub height pressure perturbation (bottom), further averaged between $y = \pm 2.5$ km. Vertical dashed lines refer to the first and lasts wind farm rows.

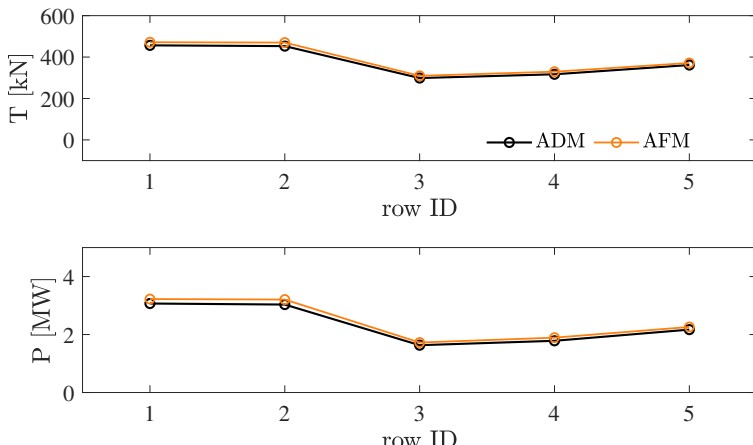

**Figure 27.** Row- and time-averaged turbine thrust (top) and power (bottom), for the two simulations presented in this section.





# 7 Conclusions

In this study, we introduced the actuator farm model (AFM), a new parametrization that allows to capture the aerodynamics of wind turbines in the context of wind farm LES. Unlike similar models such as the actuator disk (AD) or the actuator line (AL), wind turbines are represented within the AFM using a single actuator point, located at the rotor center, and only $2-3$ mesh cells are required along the rotor diameter. The turbine force is distributed to the surrounding cells by means of a new projection function whose spatial support is axisymmetric in the rotor plane and characterized by a Gaussian decay in the streamwise direction. The size of the spatial support is controlled by means of three free parameters, namely the half-decay radius on the rotor plane $r_{1/2}$, the smoothness $s$ and the streamwise standard deviation $\sigma$.

To find the best set of parameters that allow to obtain similar wake deficit profiles and turbine thrust and power to those predicted using the ADM, we conducted simulations of an isolated NREL 5MW wind turbine in uniform inflow, using different values of the horizontal grid spacing. In particular, while $\sigma$ is chosen using existing best practices from the ADM, our results show that $r_{1/2}$ should be approximately of the size of the turbine radius, while values of $s$ should lie between 6 and 10. With this choice of AFM projection parameters, results are fairly independent of the horizontal grid spacing up to a resolution of $60 \times 60$ m, i.e. $\Delta x / R \approx 2$.

The optimal set of parameters ($r_{1/2} = R$ and $s = 6$) were used to investigate the AFM performance in predicting the flow around a wind farm with 25 NREL 5MW turbines organized in 5 rows and 5 columns both in an aligned and a staggered layout, using horizontal grid spacing of $30 \times 12.5$ m, $30 \times 20$ m, $40 \times 40$ m and $60 \times 60$ m. As the wind farm is immersed in a fully neutral ABL, the time-resolved inflow condition is mapped from a previously conducted precursor simulation, where no turbines are present, using linear interpolation both in space and time.

To avoid alteration in the velocity fluctuations when the target grid is more than twice as coarse as the source grid, the precursor used to prescribe the inlet flow to the wind farm simulations with horizontal grid spacing of $40 \times 40$ m and $60 \times 60$ m has been conducted on a grid characterized by a $50 \times 50 \times 10$ m resolution. Notably, this does not satisfy the Brasseur and Wei (2010) criteria and also leads to a reduction in the predicted shear stress magnitude. To correct this issue, we proposed a modified wall model that allows to recover the shear stress profile obtained when the precursor simulation complies with the Brasseur and Wei (2010) criteria. This is achieved by prescribing the friction velocity used to compute the wall shear stress instead of calculating it based on the velocity at the first cell.

Results obtained using the AFM have been compared against ADM predictions made on the finer grid, which satisfies the ADM requirement of having at least 10 grid cells along the rotor diameters. Specifically, when the same or the $30 \times 20$ m horizontal grid spacing are employed, AFM and ADM essentially predict identical velocity and shear stress profiles around the wind farm. Moreover, row-averaged turbine thrust and power are in excellent agreement. For the $40 \times 40$ m and the $60 \times 60$ m grid spacing, the AFM captures the wind farm power at the non-waked rows, while power is under-predicted at the waked turbines. Nevertheless, all values of grid resolution allow to capture the mean velocity distribution both upstream and downstream of the wind farm with good accuracy. Therefore, for those problems where a turbine array of interest is waked by an upwind wind farm, the upwind farm may be modeled using the AFM together with a mesh resolution of $40 - 60$ m, while





the array of interest should be discretized with no less than a 30 m horizontal resolution in order to properly capture individual
wake interactions and turbine power. For comparison, this value of grid resolution would not allow to use the ADM model, as
it corresponds to only 4 cells along the rotor diameter for the wind turbine employed in the present study.

Lastly, the AFM is combined with the nested domain technique and used in two wind farm LES applications to demonstrate
its ability to drastically reduce the computational cost whilst predicting similar results in terms of flow field and turbine
variables. In particular, we conduct simulations of two aligned wind farms immersed in a truly neutral ABL and of a single
wind farm that interacts with a conventionally neutral boundary layer by triggering both internal gravity waves in the free
atmosphere and surface waves in the capping inversion layer. Conventionally, these applications are rendered computationally
intense by the large domain size required to capture the processes and the fine grid resolution imposed by the ADM model.
The proposed AFM allows to increase the grid spacing, leading to a reduced cell count. In particular, both analyses employ a
one-way coupling between an outer domain characterized by an horizontal resolution of $50 \times 50$ m and a nested inner domain
with a $30 \times 12.5$ m grid. Notably, only the solution in the inner domain is influenced by the outer domain. Hence, while the outer
domain should contain all the relevant physics, the inner domain only provides a refined solution for the region of interest. As a
consequence, turbines are modeled using the AFM and ADM in the outer and inner domain, respectively. In both applications,
the combined use of AFM and grid nesting yields velocity, shear stress and turbine quantities that are in excellent agreement
with those obtained using a finer grid and ADM throughout. Finally, we also highlight that flow perturbations induced in the
free atmosphere and within the boundary layer by wind farm-induced atmospheric gravity waves obtained using the AFM and
a coarser grid size agree almost exactly with ADM simulations conducted on a finer grid.

Future studies will involve using the AFM to study the wind farm response to more realistic atmospheric inflow conditions,
introduced within the LES using profile assimilation techniques, as well as the mutual interaction of real-world wind farms
with neighbouring clusters (off-shore) and with complex terrain features (on-shore).



*Code availability.* TOSCA is available at https://doi.org/10.17605/OSF.IO/Q4VAF (Stipa et al., 2023b).

*Data availability.* The dataset can be made available from the authors upon request.

*Author contributions.* Conceptualization, S.S., J.B.; methodology, S.S., A.A.; software, S.S., A.A.; validation, S.S.; formal analysis, S.S.; investigation, S.S.; computational resources, J.B.; data curation, S.S.; writing–original draft preparation, S.S., A.A.; writing–review and editing, J.B.; visualization, S.S.; supervision, J.B.; project administration, J.B.; funding acquisition, J.B.. All authors have read and agreed to
800 the published version of the manuscript.

*Competing interests.* The contact author has declared that none of the authors has any competing interests.

*Acknowledgements.* Computational resources provided by the Digital Research Alliance of Canada and Advanced Research Computing at the University of British Columbia are gratefully acknowledged.

*Financial support.* This research has been supported by the Natural Sciences and Engineering Research Council of Canada (grant no.
556326).





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
