# Peer review of "The Actuator Farm Model for LES of Wind Farm-Induced Atmospheric Gravity Waves and Farm-Farm Interaction"

_Wind Energy Science, 2024_

## Referee Comment (RC1)

**Review: *The Actuator Farm Model for LES of Wind Farm-Induced Atmospheric Gravity Waves and Farm-Farm Interaction**

**General Comments**

This study analyzes the performance of a new wind turbine parameterization called the actuator farm model (AFM). The authors convincingly argue that the AFM can produce similar results as the widely known actuator disk model (ADM) but at coarser resolutions. This shows that the AFM is a very useful parameterization due to the fact that LES can be run at coarser resolution to accurately simulate large wind farms which saves on computational costs. There are many applications for the AFM model – one of which is modeling wind farm induced gravity waves – and it appears to be a parameterization worth further use and development by the scientific community.

I do not see any major issues with this study and have noted a few suggestions / minor revisions below. Thus, **I recommend the article for publication with minor changes**.

**Major Revisions**

None.

**Minor Revisions**

This paper feels very long. The introduction has information that doesn't seem relevant to the topic the paper is addressing. Additionally, there are a lot of tests that are included (the gravity wave tests feel like they could be a study on their own) and it makes it difficult to retain all of the significant findings. I'm not sure what the solution is for this, but if the authors have ideas of how to make the paper more concise, I think it would benefit the paper a lot.

**Technical Suggestions**

- The abstract is quite lengthy. Consider shortening it so readers can more easily find the key message.
- Some figures (e.g., 8, 9, 16, 17, etc.) are quite small and difficult to see the aspects that are discussed in the text. Consider changing the orientation of the panels or decreasing the number of panels to make the figures more legible

---

## Referee Comment (RC2)

Review of the paper "The Actuator Farm Model for LES of Wind Farm-Induced Atmospheric Gravity Waves and Farm-Farm Interaction".

Submitted to **_Wind Energy Science_**
Article number # : wes-2024-89
August 19, 2024

**Recommendation**: Minor revision.

**Summary**:
This study introduces the actuator farm model (AFM), a novel approach for simulating wind turbines within large eddy simulations (LES). The authors show that the proposed new AFM provides wake deficit profiles, turbine thrust, and power predictions comparable to the actuator disk model (ADM), even with coarser grid resolutions. The topic is of significant interest, the methodology is robust, and the findings contribute substantially to our understanding of modeling wind farms with coarser resolutions. However, several issues must be addressed before the manuscript can be recommended for publication. My comments are categorized as either 'Major concerns' or 'Minor concerns', with the former focusing on conceptual technical critiques, and the latter highlighting grammatical and spelling errors.

**Major concerns**:

- **(1)**: The abstract mentions the introduction of a "new wall modeling approach" to ensure a correct shear stress profile, but it lacks specific details on the settings or implementation of this approach. Without further information, it's unclear how this wall modeling technique is configured or applied within the simulations. Further details in the main text would likely address these aspects and provide insight into the practical application of this new approach.

- **(2)**: The paper would benefit from explicitly mentioning any additional requirements or considerations for using the proposed AFM model. For example, it should clarify whether specific conditions need to be met, such as ensuring that the center of the turbine is aligned with a grid point when using a very coarse resolution.

**References**

---

## Author Response (AR1)

**University of British Columbia**

**The Actuator Farm Model for LES of Wind Farm-Induced Atmospheric Gravity Waves and Farm-Farm Interaction**

**Response to Reviewer 1**

Exec. S. Stipa - September 12, 2024

We would like to thank the reviewer for the time dedicated to revising the paper. We proceed with answering and clarifying, where possible, their comments.

Our response, denoted in black, is shown below, while the reviewer's comments are denoted in blue. Please refer to the track changes document for a detailed overview of the changes made to the manuscript.

This paper feels very long. The introduction has information that does not seem relevant to the topic the paper is addressing. Additionally, there are a lot of tests that are included (the gravity wave tests feel like they could be a study on their own) and it makes it difficult to retain all of the significant findings. I'm not sure what the solution is for this, but if the authors have ideas of how to make the paper more concise, I think it would benefit the paper a lot.

We agree with the reviewer that this paper can be perceived as long. For this reason, two sections of the paper have been moved to an appendix, resulting in an enhanced focus of the manuscript towards the AFM. In particular, former Sect. 2.3, which explains a correction to a wall model based on the classic Monin and Obukhov (1954) similarity theory, and former Sect. 2.4, where such wall model correction is verified for precursor simulations, have been moved to appendices A and B, respectively. This extensively shortens the paper and simplifies its structure around the sole AFM.

However, the authors would like to stress that, in general, the development of an actuator model where a coarser grid resolution can be employed with good accuracy compared to older parameterizations that employ a finer grid, cannot be used as-is in the context of wall modeled LES of the ABL. In this case, the grid coarsening allowed by using the AFM has to be corroborated by an accurate representation of the vertical velocity and shear stress profiles within the ABL. While ensuring that this happens has nothing to do with the AFM itself, it is something that always has to be considered as the grid is coarsened. Moreover, it does have an impact on the simulation results, as the ABL evolution remains negatively affected by grid coarsening when using conventional wall modeling approaches.

For this reason, we were reluctant to relegate to an appendix our simple approach that allows us to retain the vertical velocity and shear stress profile observed in finer LES simulations, as it crucially allows us to conduct wall modeled LESs of the ABL flow around a wind farm when it is modeled using the AFM. In the end, although the topic is important, it does not represent the core of the manuscript and for this reason we agree to move it to an appendix. Nevertheless, we strongly believe that it should be retained at least in this form as it crucially relates to those studies for which the AFM is most useful, i.e. the farm-farm interaction and the farm-gravity wave interaction cases.

Regarding the introduction, we feel that the story line is solid. In fact, after emphasizing the fact that large-scale installations are uncovering new physical phenomena in a broader range of spatial and temporal scales (e.g. cluster wakes and gravity waves), we highlight the various modeling tools that can be used to study this new problem today. In our opinion, it is very important at this point to address the limitations of engineering models as they (alongside the limitations of wind farm parameterizations in WRF) are what motivates large-scale LES studies for which the AFM will be useful.

Finally, we believe that the two large test cases performed in the paper (cluster wakes and gravity waves) are good examples of how the model can be directly applied for more realistic applications. This in our opinion makes the manuscript more complete and it is also beneficial for future use of the model by the research community in similar applications.

The abstract is quite lengthy. Consider shortening it so readers can more easily find the key message.

The abstract has been shortened by removing information non strictly related to the AFM and its usage exemplified by the test cases covered throughout the manuscript.

Some figures (e.g., 8, 9, 16, 17, etc.) are quite small and difficult to see the aspects that are discussed in the text. Consider changing the orientation of the panels or decreasing the number of panels to make the figures more legible.

The size of former Figs. 8, 9, 16, 22, 23 has been increased. These are now Figs. 7, 8, 15, 21, 22, respectively, in the revised manuscript.

**References**

Monin, A. and Obukhov, A.: Basic laws of turbulent mixing in the surface layer of the atmosphere, Tr. Akad. Nauk SSSR Geophiz. Inst., 151, 163–187, 1954.

University of British Columbia

The Actuator Farm Model for LES of Wind Farm-Induced Atmospheric
Gravity Waves and Farm-Farm Interaction

**Response to Reviewer 2**

Exec. S. Stipa - September 12, 2024

We would like to thank the reviewer for the time dedicated to revising the paper. We proceed with answering and clarifying, where possible, their comments.

Our response, denoted in black, is shown below, while the reviewer's comments are denoted in blue. Please refer to the track changes document for a detailed overview of the changes made to the manuscript.

The abstract mentions the introduction of a "new wall modeling approach" to ensure a correct shear stress profile, but it lacks specific details on the settings or implementation of this approach. Without further information, it's unclear how this wall modeling technique is configured or applied within the simulations. Further details in the main text would likely address these aspects and provide insight into the practical application of this new approach.

In trying to merge the requests from the two reviewers, we decided to move former Sect. 2.3, which explains a correction to a wall model based on the classic Monin and Obukhov (1954) similarity theory, and former Sect. 2.4, where such wall model correction is verified for precursor simulations, to appendices A and B, respectively. In fact, the presentation of what is more of a correction to classic wall modeling formulations than a new wall model itself is a minor finding of the present manuscript. Notably, we highlight that a wall model correction when coarsening the grid goes hand in hand with the adoption of the AFM in more realistic cases where e.g. the time-varying inflow condition that characterizes the ABL is considered, as such grid coarsening is the basic motivation for which the AFM is employed in the first place. We also note that such discussion does not apply to more idealized cases such as e.g. simulations characterized by a uniform inflow.

The approach is now extensively documented in Appendix A and verified in Appendix B on the precursor simulations used throughout the manuscript. By moving these parts to appendices, we aim to focus the paper on the AFM, which is the main topic of the paper. Notably, we also removed from the abstract the sentence mentioned by the reviewer.

The paper would benefit from explicitly mentioning any additional requirements or considerations for using the proposed AFM model. For example, it should clarify whether specific conditions need to be met, such as ensuring that the center of the turbine is aligned with a grid point when using a very coarse resolution.

The suggested values for the parameters $r_{1/2}$, $s$ and $\sigma$ are summarized both in the isolated wind turbine section and in the conclusion. The analysis suggested by the reviewer (i.e. sensitivity of the AFM to turbine misalignment with the grid) has been performed on the coarsest grid resolution of $60 \times 60$ m and included in Appendix C of the revised manuscript. In particular, it is shown that shifting the turbine center relative to the grid results in power and thrust fluctuations which fall within 5% of the magnitude of the same variables. This is less than the fluctuations obtained when using poorly set model parameters $r_{1/2}$, $s$ and $\sigma$, as shown in Fig. 4 of the revised manuscript. We hope this strengthens the paper in a manner that addresses the reviewer's comment

**References**

Monin, A. and Obukhov, A.: Basic laws of turbulent mixing in the surface layer of the atmosphere, Tr. Akad. Nauk SSSR Geophiz. Inst., 151, 163–187, 1954.

---

## Referee Report (RR1)

Review of the paper "The Actuator Farm Model for LES of Wind Farm-Induced Atmospheric Gravity Waves and Farm-Farm Interaction (Revised version 2)".

Submitted to **_Wind Energy Science_**
Article number # : wes-2024-89-R2
September 29, 2024

**Recommendation**: Accept.

**Summary**:
The author has thoroughly addressed the comments and concerns, significantly improving the paper. The reorganization of former sections 2.3 and 2.4, along with the detailed explanation of the wall modeling approach in Appendix A and its verification in Appendix B, has made the discussion much clearer and more complete. Additionally, the newly added simulation using the coarsest grid resolution of $60{\times}60$ m, as presented in Appendix C, effectively answers my earlier questions. Overall, I believe the paper is suitable for publication in Wind Energy Science.

**References**